# LPT: Long-tailed Prompt Tuning for Image Classification

**Bowen Dong**[1]  **Pan Zhou**[2]  **Shuicheng Yan**[2]  **Wangmeng Zuo**[1,3✉]
[1]Harbin Institute of Technology  [2]National University of Singapore  [3] Peng Cheng Laboratory
{cndongsky,panzhou3,shuicheng.yan}@gmail.com, wmzuo@hit.edu.cn

## ABSTRACT

For long-tailed classification tasks, most works often pretrain a big model on a large-scale (unlabeled) dataset, and then fine-tune the whole pretrained model for adapting to long-tailed data. Though promising, fine-tuning the whole pretrained model tends to suffer from high cost in computation and deployment of different models for different tasks, as well as weakened generalization capability for overfitting to certain features of long-tailed data. To alleviate these issues, we propose an effective Long-tailed Prompt Tuning (LPT) method for long-tailed classification tasks. LPT introduces several trainable prompts into a frozen pretrained model to adapt it to long-tailed data. For better effectiveness, we divide prompts into two groups: 1) a shared prompt for the whole long-tailed dataset to learn general features and to adapt a pretrained model into the target long-tailed domain; and 2) group-specific prompts to gather group-specific features for the samples which have similar features and also to empower the pretrained model with fine-grained discrimination ability. Then we design a two-phase training paradigm to learn these prompts. In the first phase, we train the shared prompt via conventional supervised prompt tuning to adapt a pretrained model to the desired long-tailed domain. In the second phase, we use the learnt shared prompt as query to select a small best matched set for a group of similar samples from the group-specific prompt set to dig the common features of these similar samples, and then optimize these prompts with a dual sampling strategy and the asymmetric Gaussian Clouded Logit loss. By only fine-tuning a few prompts while fixing the pretrained model, LPT can reduce training cost and deployment cost by storing a few prompts, and enjoys a strong generalization ability of the pretrained model. Experiments show that on various long-tailed benchmarks, with only ∼1.1% extra trainable parameters, LPT achieves comparable or higher performance than previous whole model fine-tuning methods, and is more robust to domain-shift. Code is publicly available at https://github.com/DongSky/LPT.

## 1 INTRODUCTION

Learning from long-tailed data (Cui et al., 2019; Kang et al., 2020; Zhang et al., 2021b) is very challenging in the deep learning era, since networks often excessively overfit to majority classes while ignoring the minority classes due to the overwhelming training sample number of majority classes. To eliminate this negative effect, previous methods focus on three individual aspects: **1)** re-sampling the long-tailed data distribution (Kang et al., 2020; Li et al., 2022; 2021a; Ren et al., 2020) to achieve balance among all classes in each minibatch data, **2)** re-weighting the training loss (Cui et al., 2019; Li et al., 2022; Menon et al., 2021) to give heavier weights to minority classes, and **3)** specially-designed decoupled training (Kang et al., 2020), knowledge distillation (Li et al., 2021b) or ensemble learning (Zhou et al., 2020; Wang et al., 2020).

Although alleviating the negative effect in long-tailed learning in some sense and achieving better overall performance, these methods generally need to train both feature extractors and linear classifiers from scratch or from pretrained models on large-scale datasets, *e.g.* ImageNet (Deng et al., 2009), thus suffering from three issues. Firstly, to adapt to long-tailed data, this whole model fine-tuning requires much higher extra training cost. Secondly, fine-tuning whole model also impairs the generalization ability of the pretrained model, since the pretrained model trained on a large-scale dataset often sees abundant data and enjoys strong discriminative ability to various kinds of fea-

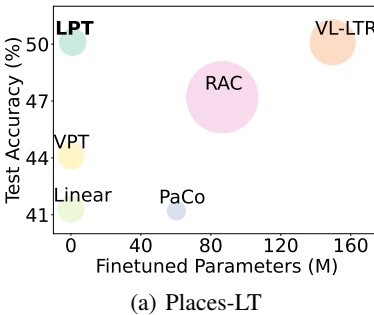 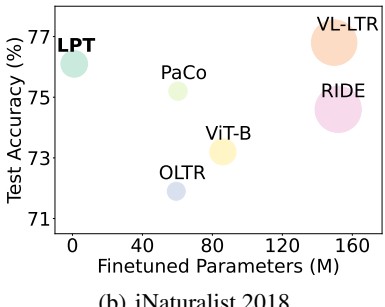

(a) Places-LT             (b) iNaturalist 2018

Figure 1: Comparison among SoTA long-tailed methods on the Places-LT dataset and iNaturalist 2018 dataset, where the size of each spot indicates the model size of the overall network, including model backbone, classier and prompts. Our LPT only needs ~1.1% additional trainable parameters while achieving comparable or higher accuracy on two highly long-tailed datasets.

tures, while fine-tuning often weaken this generalization ability caused by the overfitting to certain features of long-tailed data and hardly handle domain-shift or out-of-distributed data which occur frequently in long-tailed learning. Finally, fine-tuning also results in very different models for different learning tasks, which destroys model compatibility and increases practical deployment cost.

**Contributions.** To alleviate the above issues, we propose a novel and effective Long-tailed Prompt Tuning (LPT) approach. Specifically, LPT builds on a pretrained model, *e.g.* vision transformer (ViT) (Dosovitskiy et al., 2021), and introduces extra trainable prompts into this pretrained model, and finally only fine-tunes these prompts for adapting the pretrained model to long-tailed data at hand. For prompts, there are two kinds, **1)** shared prompt for all classes to learn general features (knowledge) and to adapt a pretrained model into the target domain; and **2)** group-specific prompts to gather group-specific features for those samples which have similar features and also to empower the pretrained model with fine-grained distinguishing ability. For effective training, we design a two-phase training framework to learn these two kinds of prompts. In the first phase, LPT optimizes the shared prompt and a classifier on a long-tailed training dataset of interest. For this phase, its target is to **1)** adapt the pretrained model to the target domain of interest via prompt tuning, and **2)** to empower the pretained model with the trained classifier the discriminative ability to the training data which is a basic to learn the group-specific prompts. During the second phase, we learn the newly added group-specific prompt set and further fine-tune classifier used in the first phase. Specifically, given an input, LPT feeds it into the pretrained model with the learnt shared prompt, and views the output class token as the query to select a small set of matched prompts via computing the cosine similarity between query and the corresponding keys from group-specific prompt set. Next, the trainable matched group-specific prompts are introduced into the pretrained model with shared prompts to help learn class-specific attributes, and is trained by asymmetric Gaussian Clouded Logit (A-GCL) loss (Li et al., 2022) with a dual sampling strategy.

This LPT can well alleviate the above three issues in existing methods as aforementioned. For training cost, LPT only needs to fine-tune a few prompts whose size is much smaller than the pretrained model, and thus uses much less training cost than fine-tuning whole pretrained model for adaptation. As for generalization ability, LPT only fine-tunes prompt while fixing the pretraining model, and thus enjoys the strong generalization capacity of the pretrained model. On compatibility, LPT shares a pretrained model for different learning tasks, and only needs to store the small-sized prompts, largely increasing the model compatibility and reducing practical deployment cost.

As shown in Fig. 1, on various long-tailed classification benchmarks, with only ~1.1% additional parameters of prompts, LPT achieves comparable or higher performance than the previous methods which fine-tunes whole pretrained model. Especially, with only vision-based data for training and testing, LPT achieves 50.1% overall classification accuracy and 46.9% few-shot accuracy on Places-LT dataset (Zhou et al., 2017a), and makes 8.9% and 11.6% improvement over the previous methods trained on vision-only data. Besides, more experimental results shows the superiority of LPT and also its generalization and robustness on long-tailed data and also domain shifted data.

## 2   RELATED WORK

**Long-tailed Image Classification.** To tackle negative effect from the highly imbalanced data distribution, previous works mainly focus on three different aspects, *i.e.*: data re-sampling (Kang et al.,

2020; Li et al., 2021a; Ren et al., 2020), which utilizes hand-crafted samplers (Kang et al., 2020), data augmentation (Li et al., 2021a) or meta-learning-based sampler (Ren et al., 2020) to balance the training data among head and tail classes; loss re-weighting (Cui et al., 2019; Menon et al., 2021; Li et al., 2022; Jamal et al., 2020; Tan et al., 2020), which focuses on adding hand-crafted bias into the confidence scores (Menon et al., 2021; Li et al., 2022), re-scaling logits by hand-crafted weights (Cui et al., 2019; Tan et al., 2020), or meta-learning-based methods (Jamal et al., 2020); and decoupled training strategies (Kang et al., 2020; Li et al., 2021b) as well as ensemble learning methods (Zhou et al., 2020; Wang et al., 2020). Recently, some vision-language-based methods (Ma et al., 2021; Tian et al., 2022; Long et al., 2022) have been proposed, which introduce extra language data (Ma et al., 2021; Tian et al., 2022) or external database (Long et al., 2022) to generate auxiliary confidence scores during training and testing, finally fine-tuning the whole CLIP-based model on long-tailed data. Different from above methods which fully fine-tune all parameters, we aim to leverage the powerful unbiased feature representation ability of pretrained models, and construct a prompt tuning method to obtain a flexible yet accurate classifier from long-tailed data.

**Efficient Tuning.** Efficient Tuning methods (including prompt (Lester et al., 2021; Jia et al., 2022), adapter (Houlsby et al., 2019; He et al., 2022; Nie et al., 2022; Chen et al., 2022), LoRA (Hu et al., 2022) and others (Frankle et al., 2021; Touvron et al., 2022)) are designed to utilize representation ability from pretrained models, and fine-tune only a few trainable parameters to achieve better performance on downstream tasks (Zhai et al., 2019; Lin et al., 2014; Zhou et al., 2017b). In this paper we focus on prompt tuning (Zhou et al., 2022a; Jia et al., 2022; Bahng et al., 2022). Specifically, Jia et al. (2022) introduced prompt into ImageNet (Deng et al., 2009) pretrained ViT (Dosovitskiy et al., 2021); while Bahng et al. (2022) inserted the prompt on the edges of images and optimize prompts. Wang et al. (2022) also introduced prompt tuning method into continue learning framework, which used multiple learnable prompts to handle corresponding tasks. Different from above works, LPT focuses on exploring the transfer ability of prompt tuning with large-scale while highly imbalanced training data, thus achieving comparable and accuracy.

## 3 PRELIMINARY STUDY

### 3.1 PERFORMANCE INVESTIGATION OF VPT

Prompt tuning in previous study (Zhou et al., 2022a; Jia et al., 2022) focuses on fine-tuning with limited data from balanced distribution (Zhai et al., 2019), while its transfer learning ability on large-scale long-tailed data (Zhou et al., 2017a; Van Horn et al., 2018) is not explored. To start our method, we first quantitatively evaluate whether prompt tuning benefits long-tailed learning or not. To this end, we investigate ViT-B (Dosovitskiy et al., 2021) pretrained on ImageNet-21k (Deng et al., 2009) by comparing the performance of linear probing and a prompt tuning method, *i.e.* VPT (Jia et al., 2022) because of its effectiveness, on the large-scale Places-LT dataset (Zhou et al., 2017a). Specifically, linear probing aims to fine-tune a linear classifier at the top of a pretrained and fixed feature extractor (*e.g.*, ViT (Dosovitskiy et al., 2021)); while VPT often concatenates input tokens with learnable prompts (tokens) and a linear classifier at top of a pretrained model. During training, we use these two methods to independently optimize their learnable parameters for 20 epochs with well-tuned hyper-parameters, *e.g.*, SGD with learning rate of 0.02 and weight decay of 1e-4.

Table 1 summarizes the quantitative results of linear probing and VPT. Without class-balanced sampling, VPT achieves 37.52% overall accuracy and surpasses linear probing by 3.94%, 3.33%, 4.52% in terms of many/medium/few-shot accuracy, respectively. Especially, after introducing class-balanced sampling (Kang et al., 2020) which first randomly sample classes from training set then randomly sample inputs with equal numbers in each iteration, VPT achieves 44.17% overall accuracy and even surpasses the counterpart by 8.67% in terms of few-shot accuracy. Based on the observation, we conclude that: **a)** prompt tuning can consistently improve the overall performance in long-tailed classification; and **b)** prompt tuning are more robust to long-tailed distribution and benefits more to tail categories. However, from Table 1, one can also observe that the performance of prompt tuning on long-tailed problem is not sufficient and yet far behind state-of-the-arts.

### 3.2 ANALYSIS OF PROMPT TUNING

Nevertheless, the reason why prompt tuning improves the performance on long-tailed learning tasks is still unclear. To quantitatively and qualitatively analyze prompt tuning, we conduct a series of experiments on Places-LT (Zhou et al., 2017a). We first adopt Linear Discriminant Analysis (LDA)

Table 1: Prompt tuning results on Places-LT (Zhou et al., 2017a). Prompt tuning achieves better accuracy on all classes and tail classes (*i.e.* "Few" in the table) with different training settings.

| Method | Balanced Sampling | Tuned Params (w/o classifier) | Overall | Many | Medium | Few |
|--------|-------------------|-------------------------------|---------|------|--------|-----|
| Linear | - | 0 | 33.29% | 46.48% | 29.45% | 18.77% |
| VPT | - | 92K | **37.52%** | **50.42%** | **32.78%** | **23.29%** |
| Linear | ✓ | 0 | 41.33% | **49.47%** | 41.31% | 27.51% |
| VPT | ✓ | 92K | **44.17%** | 45.79% | **46.73%** | **36.18%** |

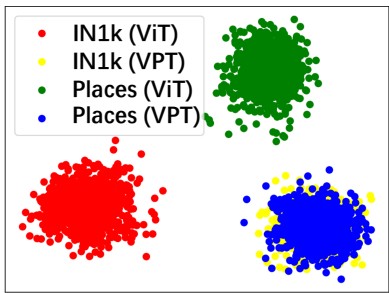

- IN1k (ViT)
- IN1k (VPT)
- Places (ViT)
- Places (VPT)

Figure 2: LDA visualization of VPT.

Table 2: Quantitative analysis of features learned by pretrained ViT-B and VPT. Features from VPT obtain better discriminative ability in terms of cluster compactness and also KNN accuracy.

| Method | ViT-B | VPT |
|--------|-------|-----|
| **Pretrain Data** | IN21k | IN21k |
| **Fine-tuned** | - | ✓ |
| **Inner-class distance $R_i$** | 2.36±0.52 | **1.82±0.43** |
| **Inner-class / inter-class $\gamma$** | 0.171 | **0.128** |
| **K-NN Acc** | 30.80% | **31.90%** |

to investigate the learned prompt from domain adaptation perspective. Specifically, we use the pretrained ViT-B and the ViT-B fine-tuned by VPT on Places-LT in Sec. 3.1 to extract features of ImageNet *val* set and Places-LT *val* set, and then employ the above features to obtain corresponding LDA vectors for visualization. From the qualitative result in Fig. 2, one can easily find that **a)** for pretrained ViT-B, its extracted features from ImageNet (red cluster) are far from its features from Places-LT (green cluster); **b)** for VPT fine-tuned ViT-B, its extracted features from ImageNet (yellow cluster) align with its features from Places-LT (blue cluster) and are close to each other. So these observations indicate that **1)** *the learned prompts in VPT could help align the fine-tuned data distribution (Places-LT) with the pretrained data distribution (ImageNet), and thus can make the pretrained model adapt to the target domain for the long-tailed learning tasks.*

Next we investigate the learned prompt from group-specific perspective. Specifically, for each class in Places-LT, we treat samples in this class as a group (cluster); then for each group i ($1 \leq i \leq C$ with total C classes in dataset), we calculate average distance between each sample and its corresponding group center, and views this average distance as inner-class distance $R_i$ of each group. Furthermore, we also define the inter-class distance $D$ as the average distance between any two group centers, and then calculate the ratio $\gamma$ between the average of inner-class distance $R_i$ and the inter-class distance $D$, namely, $\gamma = \frac{1}{CD} \sum_i R_i$. Intuitively, for a group, the smaller inner-class distance $R_i$, the more compact of the group. So if $\gamma$ is smaller, then the groups are more discriminable. Thus, we use $\gamma$ as a metric to measure whether the learnt features are distinguishable, and report the statistic results in Table 2. One can observe that features from VPT fine-tuned pretrained model achieves smaller average inner-class distance and also smaller ratio $\gamma$ than those in the vanilla pretrained model, indicating that features of different classes in VPT are easier to be distinguished. Moreover, we also conduct K-NN evaluation between the pretrained ViT-B and VPT fine-tuned pretrained ViT-B. Table 2 shows that VPT surpasses vanilla pretrained ViT-B by 1.1% in terms of K-NN accuracy, indicating the higher discriminative ability of a VPT fine-tuned model. Therefore, one can conclude that **2)** *the learned prompt can further improve the discriminative ability of pretrained models, thus benefiting to long-tailed classification problems.*

## 4 LONG-TAILED PROMPT TUNING

The observations in Sec. 3 inspire us to design an efficient yet effective long-tailed learning method based on prompt tuning (Jia et al., 2022). Nevertheless, vanilla VPT in long-tailed learning still lags behind the state-of-the-art methods (Tian et al., 2022; Long et al., 2022). To further improve the overall performance of prompt tuning in long-tailed learning, we propose an effective *Long-tailed Prompt Tuning (LPT)* method, whose framework and training procedure are illustrated in Fig. 3. Generally, LPT includes a shared prompt for all classes to learn general features or knowledge and to adapt a pretrained model into the target domain meanwhile empowering the discriminative ability to the training data; and group-specific prompts to gather group-specific features and to further fine-

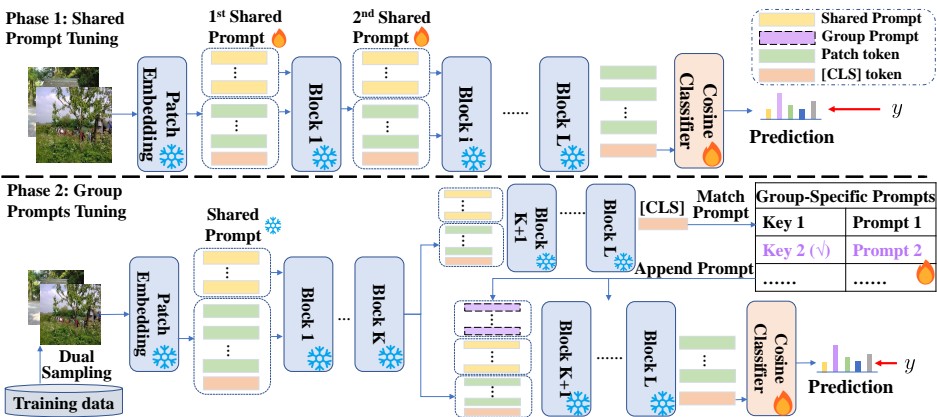

Figure 3: Pipeline of Long-tailed Prompt Tuning, where **snow** means freezed parameters and **fire** means trainable parameters. For Phase 1, LPT learns shared prompt to capture general knowledge for all classes. For Phase 2, LPT uses fixed shared prompt with ViT to generate query, then select best matched prompt from group-specific prompts, finally adopts prompt in the last L-K blocks.

tune classifier used in the first phase for higher performance. Two sets of prompts are optimized by shared prompt tuning and group prompt tuning, respectively. We introduce our LPT as follows.

## 4.1 PHASE 1: SHARED PROMPT TUNING

For Shared Prompt Tuning phase in Fig. 3, with given pretrained ViT (Dosovitskiy et al., 2021) with L layers, we aim to optimize the shared prompt $\mathbf{u} = [\mathbf{u}_1, \ldots, \mathbf{u}_L]$ and cosine classifier $f(\cdot; \theta_f)$, where $\mathbf{u}$ follows VPT-Deep (Jia et al., 2022) and consists of L individual learnable token sequences. Specifically, with given input image $\mathbf{I}$, LPT obtains the initial patch tokens $\mathbf{z}_0$ via the pretrained patch embedding layer. Then, with given class token ([CLS]) $\mathbf{c}_0$ and pretrained transformer encoder, for the i-th layer in ViT, where $1 \leq i \leq L$, we define the query used in i-th block as $\mathbf{q}_i^{\text{attn}} = [\mathbf{c}_{i-1}, \mathbf{z}_{i-1}]$, and corresponding key and value $\mathbf{k}_i^{\text{attn}} = \mathbf{v}_i^{\text{attn}} = [\mathbf{c}_{i-1}, \mathbf{z}_{i-1}, \mathbf{u}_i]$, then update $(\mathbf{c}_i, \mathbf{z}_i)$ with $\mathbf{u}$ by,

$$(\mathbf{c}_i, \mathbf{z}_i) = \text{FFN}_i(\text{Attn}_i(\mathbf{q}_i^{\text{attn}}, \mathbf{k}_i^{\text{attn}}, \mathbf{v}_i^{\text{attn}})), \tag{1}$$

where $[\cdot, \ldots, \cdot]$ denotes a token concatenation operation along the token number direction, $\text{Attn}_i$ and $\text{FFN}_i$ are the self-attention layer and feed-forward network in the i-th pretrained ViT block (Vaswani et al., 2017). Then, the final class token $\mathbf{c}_L$ are fed into the cosine classifier $f$ to calculate per-class confidence scores $\mathbf{s} = f(\mathbf{c}_L; \theta_f)$. Finally, with given ground-truth $\mathbf{y}$ of corresponding input $\mathbf{I}$, we minimize $\mathcal{L}_{P_1} = \mathcal{L}_{\text{cls}}(\mathbf{s}, \mathbf{y})$ during the training of phase 1 to optimize $\mathbf{u}$ and $\theta_f$, where $\mathcal{L}_{\text{cls}}$ is the classification loss used in both phases and will be discussed in Sec. 4.3.

## 4.2 PHASE 2: GROUP PROMPTS TUNING

A straightforward solution to reduce the difficulty of long-tailed learning is dividing the training data into multiple groups via the similarity of features, thus sharing group-specific knowledge in each group and reducing the recognition difficulty. Based on this motivation, to gather group-specific features for those samples which have similar features and also to empower the pretrained model with fine-grained discriminative ability, we aim to use different group prompts to handle samples from different classes, thus gathering group-specific features via each group prompt, benefiting to long-tailed classification. Therefore, we introduce group-specific prompts with m individual learnable prompts $\mathcal{R} = \{(\mathbf{k}_1, \mathbf{r}^1), \ldots, (\mathbf{k}_m, \mathbf{r}^m)\}$, where $\mathbf{k}_i$ is the key of the corresponding i-th group prompt $\mathbf{r}^i$ and each $\mathbf{r}^i$ has $L - K$ trainable token sequences. To reduce the computational cost and number of additional parameters, we keep using only shared prompt in the first K blocks and introduce group-specific prompt set $\mathcal{R}$ into the last $L-K$ blocks. In this subsection, we mainly discuss the training procedure of group prompts tuning. Specifically, based on our observation (2) in Sec. 3.2, we select the query $\mathbf{q} = \mathbf{c}_L$ from Phase 1 rather than using output class token from pretrained ViT like Wang et al. (2022), since the class token $\mathbf{c}_L$ often enjoys stronger discriminative ability. Given the query $\mathbf{q}$, we select the best-matched prompts adaptively from $\mathcal{R}$ by:

$$[\mathbf{w}_1, \ldots, \mathbf{w}_k] = \text{top-k}(\langle \mathbf{q}, [\mathbf{k}_1, \ldots, \mathbf{k}_m] \rangle, k) \tag{2}$$

where top-k$(\cdot, k)$ returns the indices of prompts $\mathbf{w} = [\mathrm{w}_1, \ldots, \mathrm{w}_k]$ with the largest $k$ cosine similarities, and $\langle \cdot, \cdot \rangle$ means the cosine similarity operator. Here we discuss the optimization of keys. Intuitively, one straightforward method to optimize keys is enforcing queries from the same class to match certain keys. However, this method is infeasible since it is difficult to interpret which classes could be matched into some certain prompts exactly. Instead, we prefer to simply minimize the distance between the matched queries and keys, thus optimizing these keys adaptively. We design such query function from this perspective. As observed in Sec. 3.2, the feature cluster of each class generated by the fine-tuned phase 1 is compact. Therefore, for queries from the same class, if we randomly select a query $\mathbf{q}_i$ and a key $\mathbf{k}'$ then minimize $1 - \langle \mathbf{q}_i, \mathbf{k}' \rangle$, distance between $\mathbf{k}'$ and other queries are naturally minimized since these queries are fixed and compact enough. Thus during training, each key are learnt to close to one or multiple nearby clusters, finally guiding corresponding group prompt to gather group-specific feature. Moreover, since **1)** VPT (Jia et al., 2022) benefits from prompt ensembling, and **2)** introducing more group-specific knowledge may benefit to recognize samples for tail classes. Instead of using only one matched group prompt from $\mathcal{R}$, LPT also conduct prompt ensembling with multiple selected prompts, which is shown as:

$$\mathbf{r} = \text{sum}([\mathbf{r}^{\mathrm{w}_1}, \ldots, \mathbf{r}^{\mathrm{w}_k}])/k, \tag{3}$$

thus resulting an ensembled group prompt $\mathbf{r}$. With given $\mathbf{r}$, LPT reuses the feature $(\mathbf{c}_K, \mathbf{z}_K)$ from Phase 1 as $(\hat{\mathbf{c}}_K, \hat{\mathbf{z}}_K)$ to save computational cost, then define the query used in i-th block as $\hat{\mathbf{q}}_i^{\mathrm{attn}} = [\hat{\mathbf{c}}_{i\text{-}1}, \hat{\mathbf{z}}_{i\text{-}1}]$, and key with value $\hat{\mathbf{k}}_i^{\mathrm{attn}} = \hat{\mathbf{v}}_i^{\mathrm{attn}} = [\hat{\mathbf{c}}_{i\text{-}1}, \hat{\mathbf{z}}_{i\text{-}1}, \mathbf{u}_i, \mathbf{r}_{i\text{-}K}]$, finally update $(\hat{\mathbf{c}}_i, \hat{\mathbf{z}}_i)$ as:

$$(\hat{\mathbf{c}}_i, \hat{\mathbf{z}}_i) = \text{FFN}_i(\text{Attn}_i(\hat{\mathbf{q}}_i^{\mathrm{attn}}, \hat{\mathbf{k}}_i^{\mathrm{attn}}, \hat{\mathbf{v}}_i^{\mathrm{attn}})), \tag{4}$$

where K+1 $\leq$ i $\leq$ L indicates the index of the last L $-$ K pretrained blocks in ViT. Next, the output class token $\hat{\mathbf{c}}_L$ are fed into the cosine classifier $f$ and calculate per-class confidence scores by $\hat{\mathbf{s}} = f(\hat{\mathbf{c}}_L; \theta_f)$. Finally, with given ground-truth $\mathbf{y}$ of corresponding input $\mathbf{I}$, we minimize $\mathcal{L}_{P_2}$ including both classification loss $\mathcal{L}_{\mathrm{cls}}$ as well as the cosine similarity between query $\mathbf{q}$ and corresponding matched keys $[\mathbf{k}_{\mathrm{w}_1}, \ldots, \mathbf{k}_{\mathrm{w}_k}]$, which is shown as Eqn. 5:

$$\mathcal{L}_{P_2} = \beta \mathcal{L}_{\mathrm{cls}}(\hat{\mathbf{s}}, \mathbf{y}) + (1 - \frac{1}{k} \sum_{i \in \mathbf{w}} \langle \mathbf{q}, \mathbf{k}_i \rangle), \tag{5}$$

where $\beta$ is scale factor of $\mathcal{L}_{\mathrm{cls}}$ and will be discussed in the following.

Note that naively using class-balanced sampling (Kang et al., 2020) or instance-balanced sampling (Kang et al., 2020) may lead to severe overfitting on tail classes or head classes (Zhang et al., 2021b) respectively. To balance the performance between head classes and tail classes and avoid overfitting, we introduce dual sampling strategy. Specifically, for each training iteration in Phase 2, LPT randomly samples a mini-batch $\{\mathbf{I}\}_{\mathrm{ins}}$ from instance-balanced sampler as well as another mini-batch $\{\mathbf{I}\}_{\mathrm{bal}}$ from class-balanced sampler. For samples in $\{\mathbf{I}\}_{\mathrm{bal}}$, we simply set $\beta = 1$ to calculate $\mathcal{L}_{P_2}$; and for samples in $\{\mathbf{I}\}_{\mathrm{ins}}$, we set $\beta = \eta(E - e)/E$, where $\eta$ is the initialized weight for $\{\mathbf{I}\}_{\mathrm{ins}}$, E denotes the maximum number of epochs, and e is the current epoch number.

## 4.3 LOSS FUNCTION

Finally, we introduce the classification loss $\mathcal{L}_{\mathrm{cls}}$ used in our two-phase training. Though LPT can use multiple classification losses to further improve the performance of LPT, we adopt asymmetric GCL loss $\mathcal{L}_{\mathrm{A\text{-}GCL}}$ for both adjusting logits based on statistic label frequency from training data and re-weighting gradient between positve and negative classes. Without loss of generality, we use $\hat{\mathbf{s}} = f(\hat{\mathbf{c}}_L; \theta_f)$ calculated in the Phase 2 of LPT as example to demonstrate $\mathcal{L}_{\mathrm{A\text{-}GCL}}$. Following (Li et al., 2022), we re-scale the confidence score of i-th class by:

$$\mathbf{v}_i = \alpha(\hat{\mathbf{s}}_i - (\log n_{\max} - \log n_i) \|\epsilon\|) \tag{6}$$

where $\alpha$ is the scaling factor, $\epsilon$ is the random variable from gaussian distribution, $n_i$ and $n_{\max}$ mean the label frequency of i-th class and the maximum label frequency in the training set, respectively. Then, we calculate per-class probability $\mathbf{p} = [\mathbf{p}_1, \ldots, \mathbf{p}_C]$ by:

$$[\mathbf{p}_1, \ldots, \mathbf{p}_C] = \text{softmax}([\mathbf{v}_1, \ldots, \mathbf{v}_C]). \tag{7}$$

Next, we use asymmetric re-weighting (Ridnik et al., 2021) to eliminate the effect from negative gradient in long-tailed learning. Suppose j is ground-truth class of $\mathbf{I}$, we calculate $\mathcal{L}_{\mathrm{A\text{-}GCL}}$ as:

$$\mathcal{L}_{\mathrm{A\text{-}GCL}} = (1 - \mathbf{p}_j)^{\lambda_+} \log(\mathbf{p}_j) + \sum_{1 \leq i \leq C, i \neq j} (\mathbf{p}_i)^{\lambda_-} \log(\mathbf{p}_i), \tag{8}$$

where $\lambda_+$ and $\lambda_-$ is the focusing parameter (Lin et al., 2017) for ground-truth class and negative classes respectively. Finally, we choose $\mathcal{L}_{\mathrm{cls}} = \mathcal{L}_{\mathrm{A\text{-}GCL}}$ during two-phase training of LPT.

Table 3: Comparison with state-of-the-art long-tailed classification methods on Places-LT dataset (Zhou et al., 2017a). Our LPT achieves state-of-the-art performance among vision-only pretrained methods and achieves the same performance with state-of-the-art VL-based methods.

| Method | Backbone | Tuned Params | Total Params | Extra Data (Inference) | Overall | Many | Medium | Few |
|---|---|---|---|---|---|---|---|---|
| **Vision-only Pretrained** | | | | | | | | |
| OLTR (Liu et al., 2019) | Res152 | 60.34M | 60.34M | - | 35.9 | 44.7 | 37.0 | 25.3 |
| TADE (Zhang et al., 2021a) | Res152 | 60.34M | 60.34M | - | 38.8 | 42.8 | 39.0 | 31.2 |
| LWS (Kang et al., 2020) | Res152 | 60.34M | 60.34M | - | 37.6 | 40.6 | 39.1 | 28.6 |
| MisLAS (Zhong et al., 2021) | Res152 | 60.34M | 60.34M | - | 40.4 | 39.6 | 43.3 | 36.1 |
| ALA (Zhao et al., 2022) | Res152 | 60.34M | 60.34M | - | 40.1 | 43.9 | 40.1 | 32.9 |
| PaCo (Cui et al., 2021) | Res152 | 60.34M | 60.34M | - | 41.2 | 36.1 | 47.9 | 35.3 |
| VPT (Jia et al., 2022) | ViT-B | **0.09M** | 86.66M | - | 37.5 | **50.4** | 33.8 | 23.3 |
| LPT (Ours) | ViT-B | **1.01M** | 87.58M | - | **50.1** | 49.3 | **52.3** | **46.9** |
| **Vision-Languge Pretrained with Extra Data** | | | | | | | | |
| RAC (Long et al., 2022) | ViT-B | 86.57M | 236.19M | IN21k Feat | 47.2 | 48.7 | 48.3 | 41.8 |
| BALLAD (Ma et al., 2021) | ViT-B | 149.62M | 149.62M | - | 49.5 | 49.3 | **50.2** | **48.4** |
| VL-LTR (Tian et al., 2022) | ViT-B | 149.62M | 149.62M | Wiki Text | 50.1 | **54.2** | 48.5 | 42.0 |

Table 4: Comparison with state-of-the-art methods on CIFAR100-LT with various imbalanced ratio $\tau$. LPT performs best among all methods.

| Imb Ratio $\tau$ | 200 | 100 | 50 | 10 |
|---|---|---|---|---|
| **Training from Scratch** | | | | |
| PaCo | - | 52.0 | 56.0 | 64.2 |
| Zhu et al. (2022) | - | 51.9 | 56.6 | 64.9 |
| Li et al. (2022) | 44.9 | 48.7 | 53.6 | - |
| **Vision-only Pretrained** | | | | |
| VPT | 72.8 | 81.0 | 84.8 | 89.6 |
| LPT (Ours) | **87.9** | **89.1** | **90.0** | **91.0** |
| **VL Pretrained with Extra Data** | | | | |
| Ma et al. (2021) | - | 77.8 | - | - |

Table 5: Comparison with state-of-the-art methods on iNaturalist 2018. LPT performs best among vision-only pretrained methods.

| Method | Overall | Few-shot |
|---|---|---|
| **Vision-only Pretrained** | | |
| TADE | 72.9 | - |
| PaCo | 75.2 | 74.7 |
| ViT-B/16 | 73.2 | - |
| Iscen et al. (2021) | 75.3 | 73.2 |
| ViT-L/16 | 75.9 | - |
| LPT (Ours) | **76.1** | **79.3** |
| **VL Pretrained with Extra Data** | | |
| VL-LTR | 76.8 | - |
| RAC | **80.2** | **81.0** |

## 5 EXPERIMENTS

### 5.1 COMPARISON WITH STATE-OF-THE-ART METHODS

Here we show essential comparison and analysis. More details are shown in Appendix B and C.

**Comparison on Places-LT.** Generally, these methods can be roughly divided into two groups, *i.e.*, vision-only pretrained methods and vision-language (VL) pretrained methods. Note that VL-based methods (Tian et al., 2022; Long et al., 2022; Ma et al., 2021) may introduce extra data (*i.e.*, Wiki text data or external ImageNet-21k database) during training and testing. Our LPT belongs to the first group and does not rely on extra data. Table 3 lists the evaluation results of competing methods. Compared to other vision-only pretrained methods, with **only 1.01M** (1.1%) additional trainable parameters, LPT achieves 50.1% and 46.9% in terms of overall accuracy and few-shot accuracy, and respectively surpasses the state-of-the-art PaCo (Cui et al., 2021) by 8.9% and 11.6%. Even compared with VL-LTR (Tian et al., 2022) which is a VL-based method with extra data in training and testing, our LPT achieves the same overall accuracy while obtaining higher few-shot accuracy.

**Comparison on CIFAR100-LT.** Then we evaluate LPT on CIFAR100-LT (Krizhevsky, 2009; Cui et al., 2019) with imbalanced ratio $\tau = 10, 50, 100, 200$. Evaluation results are given in Table 4. Our LPT outperforms all the competing methods on four different imbalanced ratios. Especially, LPT surpasses the CLIP-pretrained BALLAD by 11.3% in terms of accuracy with $\tau = 100$. These results indicate the effectiveness of LPT in common object-centric data with long-tailed distribution.

**Comparison on iNaturalist 2018.** Finally, we explore LPT on large-scale and fine-graind iNaturalist 2018 (Van Horn et al., 2018). Quantitative results are given in Table 5. LPT achieves 76.1% overall and 79.3% few-shot accuracy and surpasses all other state-of-the-art methods with vision-only pretrained models. Especially, LPT also surpasses fully fine-tuned ViT-L/16 (Touvron et al., 2022) by 0.2%. These results demonstrate that LPT can also handle large-scale long-tailed data with only prompt tuning and achieve comparable accuracy. Note that since VL-based methods (Tian et al., 2022; Long et al., 2022) leverage both CLIP pretrained models and extra testing data, mean-

Table 6: ImageNet-Sketch evaluation results from different fine-tuning methods.

| Method | Backbone | Overall |
|---|---|---|
| Linear Probe | ViT-B | 31.55% |
| Full fine-tune | ViT-B | 32.25% |
| LPT (Ours) | ViT-B | **36.22%** |

Table 7: Ablation study of LPT with different pretrained model sizes.

| Backbone | Phase 1 Acc | LPT Acc |
|---|---|---|
| ViT-T | 32.55% | **37.40%** |
| ViT-S | 40.50% | **44.66%** |
| ViT-B | 49.41% | **50.07%** |

Table 8: Ablation study of each phase in LPT on Places-LT benchmark (Zhou et al., 2017a).

| Method | Prompt | Phase 1 | $\mathcal{L}_{\text{A-GCL}}$ | Phase 2 | Overall | Many | Medium | Few |
|---|---|---|---|---|---|---|---|---|
| Linear | - | - | - | - | 33.29% | 46.48% | 29.45% | 18.77% |
| VPT | ✓ | - | - | - | 37.52% | **50.42%** | 33.78% | 23.29% |
| (a) | - | ✓ | - | - | 41.33% | 49.47% | 41.31% | 27.51% |
| (b) | ✓ | ✓ | - | - | 49.10% | **49.62%** | 51.53% | 43.25% |
| (c) | ✓ | ✓ | ✓ | - | **49.41%** | 46.89% | 52.54% | 47.32% |
| (d) | ✓ | ✓ | ✓ | ✓ | **50.07%** | 49.27% | 52.31% | 46.88% |

while the extra data benefits more to object-centric long-tailed scenarios, such that LPT still has small gap with these methods. How to eliminate this gap is worth to be further explored.

## 5.2 ROBUSTNESS WITH DOMAIN SHIFT

To evaluate the robustness of our LPT with domain shift or out-of-distributed data, we optimize LPT on ImageNet-LT (Deng et al., 2009; Liu et al., 2019) *train* set, and then evaluate the fine-tuned LPT on ImageNet-Sketch (Wang et al., 2019) *val* set which includes multiple sketches from 1,000 classes and far different from natural images. Here we select linear probing and fully fine-tuning from the same pretrained ViT-B as baseline methods. Evaluation results are shown as Table 6. LPT achieves 36.22% accuracy on ImageNet-Sketch dataset, surpassing linear probing as well as fully fine-tuning by 4.67% and 3.97%. One possible explanation is that our LPT gather the domain-specific knowledge during training and adapt the pretrained model to the target domain of interest via prompt tuning, thus with given input images from other domains, LPT can transfer the feature into the domain learned in LPT, then reducing the effect from domain shifting in long-tailed learning.

## 5.3 ABLATION STUDY

**Effect of Each Phase.** First we analyze the effect of each training phase and component in LPT. Table 8 demonstrates the evaluation results of per-phase ablation study. Here we select linear probing (Dosovitskiy et al., 2021) and VPT (Jia et al., 2022) as baseline methods. After training with phase 1, type (a) and type (b) surpass their corresponding baselines by 8.04% and 11.58% in terms of overall accuracy. Meanwhile, compared to type (a), after introducing prompt for fine-tuning, type (b) surpasses type (a) by 7.77% and 15.74% in terms of overall accuracy and few-shot accuracy. These results indicate that: **1)** introducing prompt for fine-tuning benefits to the overall performance and tail classes accuracy in long-tailed learning; and **2)** the proposed phase 1 of LPT can fully release the representation ability of learnable prompt, thus leading to better classification results. Furthermore, when replacing cross entropy loss in type (b) by $\mathcal{L}_{\text{A-GCL}}$, type (c) achieves 49.41% overall accuracy and obtains 4.07% improvement in terms of few-shot accuracy. Finally, after introducing the group-specific prompts as well as phase 2 in LPT, type (d) achieves 50.07% overall accuracy on Places-LT, which indicate that using different group prompts to handle different input samples can reduce the difficulty of long-tailed learning and further improve the classification performance.

**Different Model Size and Pretrained Models.** To verify the compatibility of LPT, we evaluate LPT with ViT-Tiny/Small/Base (Dosovitskiy et al., 2021), and all ViTs are pretrained with ImageNet-21k (Deng et al., 2009). Quantitative results are shown as Table 7. LPT achieves 37.40%, 44.66% and 50.07% accuracy with three pretrained models, which surpasses models with only shared prompt by 4.85%, 4.16% and 0.66%, respectively. These results demonstrate the compatibility of LPT. Besides, for ViT-T and ViT-S which has less parameters than ViT-B, the combination of shared prompt and group-specific prompts can supply abundant domain-specific knowledge and group-specific features, thus these models benefit more from LPT and obtain larger accuracy improvement. Furthermore, we also keep using ViT-B and analyze the effect of LPT on various pretrained models (Touvron et al., 2021; Caron et al., 2021; Zhou et al., 2022b; Dosovitskiy et al., 2021). Evaluation results indicate that better self-supervised learning methods and more pretraining data lead to better accuracy, *i.e.*, 50.07% after LPT. More details are shown in Appendix C.5.

Table 9: Ablation Study of Decoupled or Joint Training. Decoupled Training is better.

| Method | Epochs | Overall |
|---|---|---|
| Joint 1&2 | 80 | 47.48% |
| Decoupled 1&2 | 40+40 | **50.07%** |

Table 10: Ablation study of query function in phase 2. Using phase 1 as query is better.

| Query Func | K-NN Acc | Phase 2 Acc |
|---|---|---|
| ViT-B | 32.11% | 49.81% |
| Phase 1 | **36.16%** | **50.07%** |

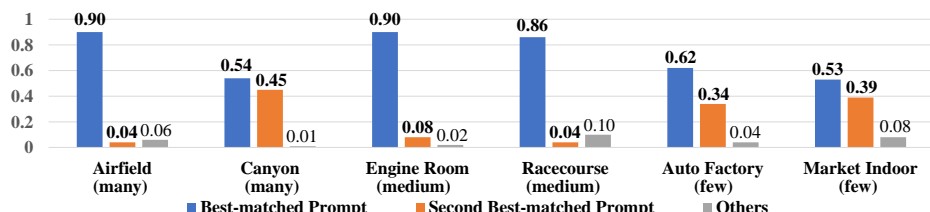

Figure 4: Statistic results visualization of prompt matching proportion for classes in Places-LT.

**Decoupled Training.** To verify the effect of decoupled training, we conduct ablation study which jointly optimizes shared prompt and group-specific prompts. For fairness, we optimize the joint-trained model with 80 epochs. The results are shown as Table 9, LPT with decoupled training achieves 50.07% overall accuracy and surpasses that with joint training by 2.59%. These results indicate that, during joint training, the shared prompt is still updated simultaneously, thus the query function is sub-optimal during training, resulting in worse matching results. Nevertheless, decoupled training leverages a fixed yet optimal shared prompt as query function, thus obtaining better results.

**Query Function and Group Size m.** We further analyze query function and group size m. Since Wang et al. (2022) used pretrained ViT as query function, to validate whether using our Phase 1 of LPT as query function is better or not, we conduct ablation study about query function. Specifically, we follow the design of LPT to use cosine similarity as distance metric and evaluate the K-NN accuracy of two query functions, then evaluate Phase 2 accuracy with different query functions, which is given in Table 10. Phase 1 achieves 36.16% K-NN accuracy and surpasses the pretrained ViT-B by 4.05%, which indicates that queries from Phase 1 obtain higher quality for matching prompts. Meanwhile, compared to LPT with ViT-B query, LPT with phase 1 query achieves 50.07% overall accuracy, which also demonstrates the effectiveness of using phase 1 as query function. Besides, we also vary the size of $\mathcal{R}$ m from 5 to 40, and find that LPT is robust to m and achieves the best accuracy 50.07% when m = 20. More details are presented in Appendix C.6.

**Statistic of Prompt Matching.** To verify that keys in group-specific prompts can adaptively learn to match samples from the same class, we count the matching results for samples in each class. And for better visualization, we randomly select two classes from many/medium/few-shot classes respectively, and then demonstrate the proportion of best-matched prompt as well as the second best-matched prompt, which is shown as Fig. 4. We notice that, for each class, samples matched by prompts with top-2 cosine similarity consists of the majority of proportion. This result is consistent with the adaptive prompt matching and prompt ensembling with $k = 2$ mentioned in Sec. 4.2, and demonstrate the effectiveness of group-specific prompts.

## 6 CONCLUSION

We proposed Long-tailed Prompt Tuning (LPT) to tackle long-tailed learning, which consists of **1)** shared prompt for all classes to learn general features or knowledge and to adapt a pretrained model into the target domain; and **2)** group-specific prompts to gather group-specific features for those samples which have similar features and also to empower the pretrained model with fine-grained discriminative ability. For effective training, we propose a two-phase training framework, in which the first phase optimizes shared prompt to adapt the pretrained model to the target domain of interest, and the second phase optimizes group-specific prompts with dual-sampling strategy and asymmetric GCL loss to dig the common features of these similar samples. Experimental results demonstrate the effectiveness and efficiency of our LPT with ~1.1% extra trainable parameters, meanwhile illustrating its robustness against domain shift.

ACKNOWLEDGEMENT

This work was supported by the National Key RD Program of China under Grant No. 2021ZD0112100.

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

The content of the Appendix is summarized as follows: 1) in Sec. A, we demonstrate the details of datasets we use in experiments of LPT; 2) in Sec. B, we demonstrate the implementation and training detail of LPT; in Sec. C, we illustrate more detailed ablation studies of LPT, including effect of shared and group prompts in C.1, pretrained models C.5, group sizes C.6, effect of K C.7, prompt ensembling C.8, dual sampling strategy C.9 and asymmetric GCL loss C.10. Meanwhile, we also illustrate more comparison results, including comparison with state-of-the-art methods with same backbone in C.2, comparison with different efficient tuning methods in C.3, comparison with multi-task learning based method in C.11 and more detailed robustness evaluation in C.4.

## A  DATASET DETAILS

**CIFAR100-LT**   is a subset of the original CIFAR-100 (Krizhevsky, 2009), which includes 100 different categories and 10,000 test images. With given imbalanced ratio $\tau = \max(n_i)/\min(n_i)$, where $1 \leq i \leq 100$ is the class index of CIFAR-100, we follow the exponential down-sampling setting from (Cao et al., 2019; Cui et al., 2019) to generate corresponding long-tailed training data, and utilize CIFAR100-LT with $\tau = 10, 50, 100, 200$ to evaluate our LPT.

**Places-LT**    (Liu et al., 2019) is the subset of Places-365 dataset Zhou et al. (2017a), which includes 62500 training images from 365 individual scene categories with per-class samples from 5 to 4980. Following previous works (Liu et al., 2019; Kang et al., 2020), we optimize our LPT on *train* set and then select the best model on *val* set, finally report the test accuracy on *test* set.

**iNaturalist 2018**    (Van Horn et al., 2018) is a large-scale fine-grained classification dataset, which exists extremely high imbalanced distribution with $\tau = 996$. This dataset includes $\sim$437K training images as well as 24.4K validation images. Following previous works (Kang et al., 2020; Cui et al., 2019), we use *train* set to optimize our LPT and evaluate LPT on *val* set.

**ImageNet-Sketch**    (Wang et al., 2022) includes 50,000 individual sketch images from the same 1,000 ImageNet categories. Since the backbone of our LPT is pretrained on ImageNet (Deng et al., 2009), different from previous methods, we only use ImageNet-LT (Liu et al., 2019) *train* set to fine-tune the prompt and classifier. Instead, we evaluate LPT on ImageNet-Sketch *val* set to verify the robustness of our LPT with domain shift scenarios.

### A.1  EVALUATION PROTOCOL

Following Cui et al. (2021); Tian et al. (2022); Long et al. (2022), for both Places-LT (Zhou et al., 2017a) and iNaturalist 2018 Van Horn et al. (2018), we first divide the whole dataset by the number of training images in each class, *i.e.*, many-shot ($\geq$100 images), medium-shot (20$\sim$100 images) and few-shot ($\leq$20 images); then we report the overall accuracy as well as many/medium/few-shot accuracy, respectively. Meanwhile, for CIFAR100-LT dataset, following Cui et al. (2021); Li et al. (2022), we directly report the overall accuracy with $\tau = 10, 50, 100, 200$. And for ImageNet-Sketch dataset, we also directly report the overall accuracy for simplicity.

## B  IMPLEMENTATION DETAILS

Following Jia et al. (2022); Wang et al. (2022), we use ViT-B/16 (Dosovitskiy et al., 2021) with ImageNet-21k pretrained model as the backbone of LPT. For shared prompt, we simply set the default length of prompt as 10 and adopt prompts on all transformer blocks in the ViT. And for group-specific prompts, we set shared layer number K = 6 and the size of prompt size m = 20; for each prompt in the set, the prompt length is also set as 10. Note that setting K = 6 may lead to 1.5x inference cost (*i.e.*, use ViT with shared prompt to generate output class token as query, then reuse features from the 6-th block and inference with both shared prompt and group prompt for the last 6 blocks) compared to VPT (Jia et al., 2022), but can achieve better accuracy, and is more efficient than inference twice (*i.e.*, inference once with pretrained ViT-B to generate the output class token as query, and then inference the second time with prompt to calculate the final confidence scores), *e.g.*, Wang et al. (2022). During training and testing, we set the prompt ensemble number $k = 2$. Finally, the number of additional parameters (linear classifiers are omitted for simplicity) is 1.01M. Both

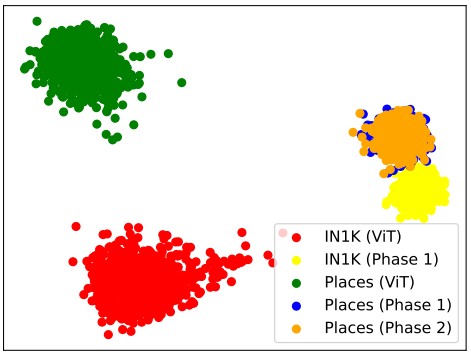

Table 13: Quantitative analysis of features learned by Phase 1 of LPT and Phase 2 of LPT. Features from phase 2 of LPT obtains the similar $\gamma$ value while obtaining larger inter-class distance.

| Method | LPT Phase 1 | LPT Phase 2 |
|---|---|---|
| **Inter-class distance** | 15.8 | **16.1** |
| **Inner-class / inter-class $\gamma$** | **0.237** | 0.242 |

Figure 5: LDA visualization of LPT.

prompts are initialized with truncated normal distribution. During training in both phase one and phase two, we use SGD optimizer with momentum of 0.9 to optimize the shared prompt and group-specific prompts, respectively. During training, the initial learning rate is set to be $0.002 * \frac{B}{256}$, where $B$ indicates batch size. Different from Alshammari et al. (2022), we set the weight decay as 1e-4 since large weight decay does not lead to significant performance improvement in LPT. During training, we use cosine learning rate scheduler to control the learning rate with 5 warmup epochs. For Places-LT, CIFAR100-LT and ImageNet-LT, we optimize phase 1 and phase 2 of LPT for E = 40 epochs, respectively; and for iNaturalist 2018, since this dataset includes much more training data, we set epoch number E = 80 for phase 1 and phase 2, respectively. For asymmetric GCL loss, we set $\lambda_+$ and $\lambda_-$ as 0 and 4, respectively. And for phase 2, we set the initialized weight $\gamma$ used in $\{\mathbf{I}\}_{\text{ins}}$ as 0.5. In all experiments, the training and testing images are resized to 224×224. During both two training phases of LPT, we only introduce random crop and resize operation and Mixup technique as data augmentation; and during evaluation, we All programs are implemented by PyTorch toolkit (Paszke et al., 2019), and all experiments are conducted on a single RTX A6000 GPU.

Table 11: Ablation Study of different pretrained models. Both more pretraining data and better pretraining algorithms benefit to LPT.

| Method | Data | Phase 1 | LPT |
|---|---|---|---|
| DINO | IN1k | 40.65% | 42.18% |
| Mugs | IN1k | 41.92% | 43.66% |
| Supervised | IN1k | 41.33% | 42.71% |
| Supervised | IN21k | **49.41%** | **50.07%** |

Table 12: Ablation Study of the size of group-specific prompts. Prompt set with 20 individual group prompts performs better.

| Group Size | Parameters | Phase 2 Acc |
|---|---|---|
| 5 | 0.23M | 49.81% |
| 10 | 0.46M | 49.90% |
| 20 | 0.92M | **50.07%** |
| 40 | 1.84M | 49.87% |

# C  MORE ABLATION STUDY AND DISCUSSION

## C.1  THE EFFECT OF SHARED PROMPT AND GROUP PROMPTS

To further investigating the effect of shared prompt and group prompts, in addition to the ablation study in Sec. 5.3, we also conduct more quantitative and qualitative analysis on Places-LT dataset. As discussed in Sec. 4, shared prompt aims to learn the domain-specific knowledge from training data while group prompts aim to gather group-specific features from training data in corresponding groups. To verify this point, we also adopt LDA among features extracted from LPT (Phase 1) and LPT (Phase 2), as well as features from pretrained ViT without any prompts. The visualization results are shown in Fig. 5. According to the visualization, we find that the distribution of feature from LPT (Phase 1) and LPT (Phase 2) are highly overlapped, this observation proves that shared prompts in phase 1 obtains the domain-specific knowledge. Moreover, we also investigate the $\gamma$ value and inter-class distance of features from LPT (Phase 1) and LPT (Phase 2), which are shown in Table 13. We find that, both of them have the similar $\gamma$, while LPT (Phase 2) has larger inter-class distance, which indicates that group prompts introduce more class discriminative ability.

Table 14: Fair comparison with state-of-the-art methods on Places-LT dataset. All methods start from the same IN21K pretrained ViT-B feature extractor to conduct fully fine-tuning or prompt tuning. Quantitative results show that LPT still largely surpasses previous methods by 7.25%.

| Method | Backbone | Pretrained Data | Overall Acc |
|---|---|---|---|
| Kang *et al.* (Kang et al., 2020) | ViT-B | IN21K | 40.45% |
| PaCo (Cui et al., 2021) | ViT-B | IN21K | 37.00% |
| GCL (Li et al., 2022) | ViT-B | IN21K | 42.82% |
| LPT | ViT-B | IN21K | 50.07% |

Table 15: Fair comparison with different efficient tuning methods on Places-LT dataset. All methods start from the same IN21K pretrained ViT-B feature extractor. Quantitative results show that LPT achieves the best accuracy.

| Method | Backbone | Pretrained Data | Overall Acc |
|---|---|---|---|
| VPT (Jia et al., 2022) | ViT-B | IN21K | 44.2% |
| Visual Prompting (Bahng et al., 2022) | ViT-B | IN21K | 13.8% |
| Pro-Tuning (Nie et al., 2022) | ViT-B | IN21K | 48.3% |
| LPT | ViT-B | IN21K | **50.1%** |
| LPT+FT-Norm (Frankle et al., 2021) | ViT-B | IN21K | 49.9% |
| LPT+Adapter (Nie et al., 2022) | ViT-B | IN21K | 49.9% |

## C.2 FAIR COMPARISON WITH STATE-OF-THE-ART METHODS

Since previous methods mainly leverage IN1k pretrained ResNet to conduct experiments on long-tailed datasets, to fairly compare LPT with previous methods via using the same backbone, we reimplement three state-of-the-art long-tailed methods (Kang et al., 2020; Cui et al., 2021; Li et al., 2022) with the same IN21K pretrained ViT-B as feature extractor to conduct fully fine-tuning on Places-LT. Experimental results are shown in Table 14. The state-of-the-art GCL (Li et al., 2022) with IN21K pretrained ViT-B backbone achieves 42.82% overall accuracy on Places-LT, while still lower than LPT by 7.25%. These results also indicate that LPT surpasses previous state-of-the-art methods under the fair comparison settings.

## C.3 COMPARISON WITH DIFFERENT EFFICIENT TUNING METHODS

Without loss of generality, we conduct the comparison between LPT and more efficient tuning methods on Places-LT dataset, including different prompt tuning methods (Visual Prompting (Bahng et al., 2022) and VPT (Jia et al., 2022)) and adapter tuning methods (e.g., Pro-Tuning (Nie et al., 2022)). Furthermore, we also additionally adopt two efficient tuning methods, i.e., normalization tuning and adapter tuning into our LPT (i.e., LPT+FT-Norm (Frankle et al., 2021) and LPT+Adapter (Nie et al., 2022)). Note that for previous efficient tuning methods, we adopt balanced sampling and use grid search to find the best training parameters to optimize the models. Without loss of generality, we keep using the same IN21K pretrained ViT as backbone for all methods and evaluate all methods on Places-LT dataset. The corresponding evaluation results are shown in Table 15. Compared to other efficient tuning methods, our LPT surpasses the state-of-the-art VPT and Pro-Tuning by 1.8% and 5.9%, respectively. These results demonstrate the effectiveness of LPT in handling long-tailed classification. And note that Bahng et al. (2022) only inserts learnable prompt on the edges of input images, and fixes all the other pretrained parameters (including the linear classifier), which may explain its relatively lower performance. And meanwhile, after adopting other efficient tuning methods into LPT, the final performance has no significant improvement. Thus we choose the current LPT structure as the final framework.

## C.4 MORE DETAILED EVALUATION FOR ROBUSTNESS WITH DOMAIN SHIFT

To further compare our LPT with the baseline methods (linear probe, fully fine-tune, VPT (Jia et al., 2022), WISE-FT (Wortsman et al., 2022)), we evaluate all these models on six different OOD

Table 16: Full comparison with different fine-tuning methods on six different OOD dataset. All methods start from the same IN21K pretrained ViT-B feature extractor. Quantitative results show that LPT achieves the best accuracy.

| Method | ImageNet-Sketch | ImageNet-ReaL | ImageNet-V2 | ImageNet-A | ImageNet-R | ObjectNet |
|---|---|---|---|---|---|---|
| Linear Probe | 31.55% | 81.43% | 63.54% | 29.20% | 45.72% | 6.61% |
| Fully Fine-tune | 32.25% | 80.10% | 62.31% | 30.12% | 43.14% | 7.13% |
| VPT | 34.64% | 85.82% | 68.51% | 35.17% | 47.06% | 8.03% |
| WISE-FT | 34.79% | 82.20% | 65.76% | 36.75% | 47.32% | 8.00% |
| LPT | **36.22%** | **87.22%** | **70.71%** | **39.65%** | **50.47%** | **8.22%** |

Table 17: Ablation Study of the number of blocks with only shared blocks, *i.e.*, K. Quantitative results show that K = 6 performs better for phase 2 of LPT.

| K | 8 | 6 | 4 |
|---|---|---|---|
| **Param Num** | 0.61M | 0.92M | 1.23M |
| **Overall Acc** | 49.77% | **50.07%** | 49.92% |

datasets (ImageNet-Sketch (Wang et al., 2019), ImageNet-ReaL (Beyer et al., 2020), ImageNet-V2 (Recht et al., 2019), ImageNet-A (Hendrycks et al., 2021b), ImageNet-R (Hendrycks et al., 2021a), ObjectNet (Barbu et al., 2019)). For fairness, all these methods start from the same IN21K pretrained ViT-B and fine-tune on the same ImageNet-LT training set. The evaluation results are shown in Table 16. Our LPT surpasses all baselines on six OOD datasets. Note that Herrmann et al. (2022) mentioned that the pretrained ViT (Dosovitskiy et al., 2021) we used in this experiment performs bad on ObjectNet benchmark (i.e., 17.36% accuracy after ImageNet-1k training), all corresponding results on ObjectNet in the table are relatively low because the training is performed on the subset of ImageNet-1k (i.e., ImageNet-LT).

## C.5 PRETRAINED MODELS

We also evaluate our LPT on various pretrained models from different pretraining algorithms and different pretraining data scale. Specifically, we keep using ViT-B/16 structure and select DeiT (Touvron et al., 2021), DINO (Caron et al., 2021) and Mugs (Zhou et al., 2022b) pretrained on ImageNet-1k (Deng et al., 2009), meanwhile using ViT-B (Dosovitskiy et al., 2021) pretrained on ImageNet-21k as baseline model. Evaluation results are shown as Table 11. With the same ImageNet-1k pretraining data, LPT with Mugs achieves 41.92% accuracy after phase 1 and 43.66% accuracy after phase 2, which surpasses other two pretrained models. Meanwhile, VPT with ImageNet-21k pretrained model achieves 50.07% accuracy and largely surpasses LPT with DeiT-B. These results indicate that pretrained models from better algorithms or larger pretraining data lead to better efficient tuning results with long-tailed target data.

## C.6 SIZE OF GROUP-SPECIFIC PROMPTS

We also conduct ablation study about the size of group-specific prompts. Generally, we start each phase 2 training from the same phase 1 model, and only change the group size of the group-specific prompts. The corresponding evaluation results are shown as Table 12. When the size of group-specific prompts increasing to 20, the accuracy of LPT increases from 49.81% to 50.07%. However, when we further increase the size to 40, the final accuracy declines to 49.87%. A possible reason is that, some classes in the dataset may share some similar group-specific feature or knowledge, such that features from instances in corresponding classes may be similar. Thus instances from these classes can be seen as a cluster and can be matched into the same prompt. Since the number of different attributes is limited, we only need a fixed number (*e.g.*, 20) of group prompts to handle the whole dataset and achieve better accuracy. Besides, using too many group prompts may increase the difficulty of clustering and optimization, which affects the final accuracy.

Table 18: Ablation Study of ensemble prompt number $k$. Quantitative results show that $k = 2$ and 3 achieve better results, thus we select $t = 2$ in LPT.

| Ensemble Num $k$ | 1 | 2 | 3 | 4 |
|---|---|---|---|---|
| Overall Acc | 49.93% | **50.07%** | 50.00% | 49.93% |
| Few-shot Acc | 46.32% | **46.88%** | 46.87% | 46.84% |

Table 19: Ablation Study of dual sampling in phase 2. Adding dual sampling strategy with proper small $\gamma$ (*e.g.*, 0.5).

| Method | Dual | $\gamma$ | Overall | Many-shot |
|---|---|---|---|---|
| (a) balanced sampling | - | - | 49.90% | 47.67% |
| (b) dual w/ small $\gamma$ | ✓ | 0.5 | **50.07%** | 49.27% |
| (c) dual w/ large $\gamma$ | ✓ | 1.0 | 49.62% | **50.06%** |

## C.7 EFFECT OF K

Next we further analyze the effect of K, which stands for the number of blocks with only the shared prompt **u**. Intuitively, the less $K$ means more parameters are inserted into more transformer blocks to conduct group prompts tuning and should lead to better performance. To verify this hypothesis, we set K $= 4/6/8$ and conduct corresponding ablation study. Evaluation results are shown in Table 17. LPT with K $= 6$ achieves the best performance and surpasses LPT with K $= 8$ by 0.3% in terms of overall accuracy. Meanwhile, LPT with K $= 4$ achieves similar performance with K $= 6$ counterpart, but does not lead to significant performance improvement with more parameters. The possible reason is that: **1)** too large K could restrict the group-specific knowledge gathering ability of group-specific prompts, since only a few layers are utilized to extract group-specific features from long-tailed data; and **2)** compared to $K = 4$, LPT with K $= 6$ fully leverages the adapted feature representation from phase 1, thus reducing the difficulty of optimization and achieving better accuracy. Therefore, we choose K $= 6$ for final LPT during experiments.

## C.8 EFFECT OF ENSEMBLE NUMBER $k$

We also analyze the effect of ensemble token number $k$ in phase 2. Intuitively, introducing can lead to better accuracy for tail classes since more class-specific knowledge are utilized for recognition. Therefore we set $k = 1/2/3/4$ and conduct corresponding ablation study. Evaluation results are shown as Table 18. Based on the results, we find that: **1)** the overall accuracy are robust with different $k$ values, and **2)** introducing prompt ensembling benefits to tail classes (+0.56% in terms of few-shot accuracy), meanwhile using top-2 best matched prompts for ensembling achieves the best results. Therefore, we choose $k = 2$ during training and testing.

## C.9 EFFECT OF DUAL SAMPLING IN PHASE 2

To evaluate the effect of dual sampling in phase 2, we conduct a series of experiments, which is shown as Table 19. Compared to type (a), type (b) achieves ∼0.2% accuracy improvement in terms of overall accuracy, meanwhile surpasses 1.6% improvement in terms of many-shot accuracy. These results indicate that: introducing dual sampling strategy with proper $\gamma$ can lead to better overall performance and reduce the overfitting from balanced sampling only, but dual sampling with too large weight may lead to negative effect on overall accuracy.

Table 20: Ablation Study of asymmetric GCL Loss. Introducing gradient re-weighting into GCL Loss can further improve overall accuracy in long-tailed classification.

| Loss Function | Overall | Many | Medium | Few |
|---|---|---|---|---|
| GCL Li et al. (2022) | 49.58% | 48.19% | 52.62% | 45.75% |
| $\mathcal{L}_{\text{A-GCL}}$ | 50.07% | 49.27% | 52.31% | 46.88% |

Table 21: Fair comparison with multi-task learning methods on Places-LT dataset. All methods start from the same IN21K pretrained ViT-B feature extractor to conduct fully fine-tuning or prompt tuning. Quantitative results show that LPT still surpasses multi-task learning method by a large margin.

| Method | Backbone | Pretrained Data | Places-LT Acc | CIFAR100-LT (IF=200) Acc |
|---|---|---|---|---|
| Multi-task | ViT-B | IN21K | 40.45% | |
| LPT (Places-LT) | ViT-B | IN21K | 50.1% | - |
| LPT (CIFAR100-LT, IF=200) | ViT-B | IN21K | - | 87.9% |

### C.10    EFFECT OF ASYMMETRIC GCL LOSS

To evaluate the effect of adding asymmetric gradient re-weighting design into GCL loss, we conduct ablation study between $\mathcal{L}_{\text{A-GCL}}$ and standard GCL loss Li et al. (2022). Without loss of generality, we conduct both experiments on phase 2. The quantitative results are shown as Table 20, LPT with $\mathcal{L}_{\text{A-GCL}}$ surpasses the counterpart with GCL loss by 0.49% and 1.03% in terms of overall accuracy and few-shot accuracy. These results further demonstrate the effect of $\mathcal{L}_{\text{A-GCL}}$.

### C.11    COMPARISON WITH MULTI-TASK LEARNING METHOD

Without loss of generality, we compare our LPT with a multi-task training based method from IN21K pretrained ViT-B to optimize both Places-LT and CIFAR100-LT with imbalanced factor of 200 (IF=200). Generally, in the multi-task training method, for each task (corresponding to a specific dataset), we initialize a linear classifier, and then optimizing all linear classifiers and the pretrained backbone by end-to-end fine-tuning. The corresponding results are shown in Table 21. LPT surpasses multi-task training method by 10.6% on Places-LT and 14.8% on CIFAR100-LT (IF=200). Benefiting from the two merits mentioned in the introduction, LPT can achieve high performance meanwhile easy to deployment with different scenarios.

### C.12    BROADER IMPACT

LPT is based on previous large-scale pretrained models, and fine-tunes only as few extra trainable parameters to adapt to real-world long-tailed scenarios. Compared to previou methods, LPT is more efficient for saving training cost and storing additional parameters, which is economic for real-world application. However, LPT is still fully data-driven, and should be cautious with potential negative impact from biased data.

### C.13    LIMITATION

The performance of head classes from LPT is still lower than that from the VL-based state-of-the-art method (Tian et al., 2022). A possible solution is proposing a novel head-tail separation algorithm (Xu et al., 2022) to further reduce the difficulty of prompt tuning with divide-and-conquer strategy, thus improving the accuracy for both head and tail classes. This part leaves for further exploration in the future.

## D    MORE STATISTIC OF PROMPT MATCHING

To verify that keys in group-specific prompts can adaptively learn to match samples from the same class, we count the matching results for samples in each class. And for better visualization, we provide more results from many/medium/few-shot classes, and then demonstrate the proportion of best-matched prompt as well as the second best-matched prompt, which is shown as Fig. 6. We notice that, for each class, samples matched by prompts with top-2 cosine similarity consists of the majority of proportion. This result is consistent with the adaptive prompt matching and prompt ensembling with $k = 2$ mentioned in Sec. 4.2, and demonstrate the effectiveness of group-specific prompts.

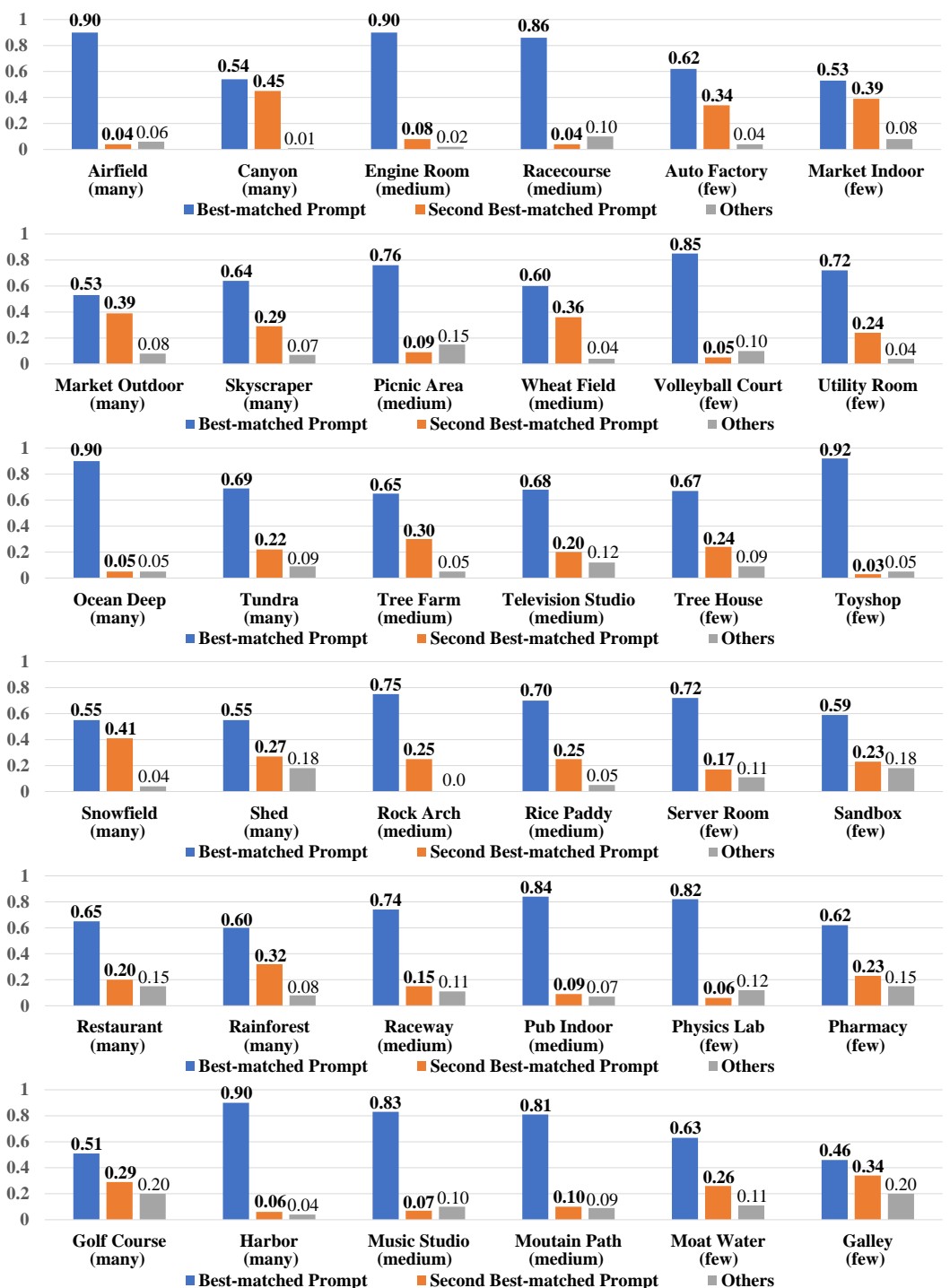

Figure 6: More statistic results visualization of prompt matching proportion for classes in Places-LT.

