# OpenReview forum: "LPT: Long-tailed Prompt Tuning  for Image Classification"
_ICLR.cc/2023/Conference — ICLR 2023 poster_

### Official Review · Reviewer_q9x4 · 2022-10-17

**Confidence:** 4
**Correctness:** 3
**Technical Novelty And Significance:** 3
**Empirical Novelty And Significance:** 3
**Recommendation:** 8

**Clarity, Quality, Novelty And Reproducibility:**

I think this paper is clear overall. Some proofreading is needed for some grammar errors. Code is not yet available.

**Strength And Weaknesses:**

The proposed method is simple yet effective and achieves general improvements over compared baselines in the reported experiments. I think it is a good contribution to the community to introduce prompt tuning to long-tailed recognition because it sparks thinking on how to effectively utilize pre-learned knowledge (pre-trained weights in this case). However, prompt tuning has a prerequisite, a "pre-trained weights" or a foundation model, which is not always approachable. And the existence of pre-trained weights shifts the goal of long-tailed methods from "balanced learning from imbalanced data" to "effectively transfer pre-trained knowledge to imbalanced data". This shift makes evaluating the model's pure recognition and generalization ability harder, and generally changes the setting of long-tail recognition. Although this might not be a bad thing, I think this shift is worth some discussion. Moreover, since the basis of the method is to solve challenges like "fine-tuning whole model impairs the generalization ability of pre-trained models", the discussion on why we need to preserve the generalization ability of pre-trained models is necessary.

**Summary Of The Paper:**

This paper introduces visual prompt tuning (VPT) methods in long-tailed recognition and proposes a modification to VPT to adapt to long-tailed scenarios with so-called long-tailed prompt tuning (LPT). The method is simple and effective. The results of compared benchmarks show improvements from the proposed LPT method.

**Summary Of The Review:**

Overall, I think this paper is good.

---

> ### Author Response · Authors · 2022-11-16
> **Author Responce to Reviewer 4**
>
> ## Q1-about "effectively transfer pre-trained knowledge to imbalanced data":
>
> Thanks for your comments. Actually, this work focuses on "effective transferring pretrained knowledge to imbalanced data". To better state our motivation of exploring efficient tuning methods in long-tailed learning, at below we first discuss the solid prerequisite of exploring efficient tuning methods, then discuss the reason why we turn from "balanced learning from imbalanced data" to "effectively transfer pretrained knowledge to imbalanced data" in long-tailed classification tasks, and finally summarize the merits of "effective transferring pretrained knowledge to imbalanced data".
>
> 1) Due to lack of 1) large-scale training data used for pretraining and 2) computation resources for pretraining, the foundation models were usually not approachable, and consequently previous works usually optimized models for downstream tasks from scratch. With the rapid development of computation resource and pretraining datasets, the community is inspired to pretrain and publish more and more foundation models of single-modality (e.g., DINO and BERT) or multi-modality (e.g., CLIP and VinVL) with various model sizes. Hence nowadays, the foundation models are widely available and approachable, and some works explore transferring pertrained knowledge to balanced data to handle corresponding downstream tasks (e.g., ViT-Adapter [1]). These models (e.g., the pretrained ViT used in this work) inspire us to investigate "effectively transferring pretrained knowledge to imbalanced data".
>
> 2) Previous long-tailed classification works mainly focus on only balanced learning from imbalanced data, i.e., transferring knowledge from many-shot classes to improve the performance to few-shot classes. However, after training, the learned feature extractors of these methods still overfit to many-shot classes, while only the learned linear classifiers benefit from balanced learning algorithms. This issue restricts the performance of long-tailed learning methods. If we replace such a  feature extractor by a fixed and powerful one, i.e. the foundation models in 1), samples from both many-shot classes and few-shot classes would benefit from pretrained knowledge, and thus the models can obtain better results with imbalanced data. That is the reason why we turn to exploring effectively transfer pretrained knowledge to imbalanced data.
>
> 3) The merits of transferring pretrained knowledge to imbalanced data are two-fold: a) benefiting from pretraining on large-scale many-shot data, pretrained models have learned a good feature representation ability, and thus transferring pretrained knowledge from imbalanced data can fully leverage the optimal feature representation ability to avoid learning imbalanced and sub-optimal feature representation from long-tailed data; and b) in real-world applications, the pretrained models are usually utilized in multiple scenarios (especially learning from long-tailed data), and thus an efficient tuning method may be easy to deploy while keeping the promising performance.
>
> These are the motivations for us to focus on this shift and to explore efficient transfer learning with imbalanced data.
>
>
> ## Q2-about "why needing to preserve the generalization ability of pretrained models is necessary"
>
> The necessity of preserving the generalization ability of pretrained models (in other words, analyzing efficient tuning methods in our work) can be summarized as follows:
>
> 1) We first discuss fully fine-tuning methods, the most widely used training strategy in long-tailed learning. As stated in the introduction section, since long-tailed downstream datasets are usually smaller and much more imbalanced than large-scale pretraining datasets, directly fine-tuning the whole pretrained models on corresponding downstream tasks will enforce the pretrained feature extractor to focus on certain features (i.e., many-shot classes in the long-tailed training data), impairing the original feature representation ability learned from large-scale training data, and finally affecting the recognition ability of few-shot classes.
>
> 2) Then we analyze the significance of preserving the generalization ability of pretrained models for efficient tuning. Preserving the generalization ability is the prerequisite of efficient tuning methods (e.g., prompt tuning methods). As stated in the introduction section, efficient tuning methods benefit from the promising generalization ability of pretrained models, and can achieve good performance with limited newly-added parameters. Thus the deployment cost of efficient tuning methods is smaller than fully fine-tuning for each downstream task.
>
> [1] Vision Transformer Adapter for Dense Predictions, 2022

---

> > ### Comment · Reviewer_q9x4 · 2022-11-23
> > **Response to rebuttle**
> >
> > Thanks for the responses. I think the additional information answers most of my questions.

---

### Official Review · Reviewer_mA6r · 2022-10-24

**Confidence:** 3
**Correctness:** 3
**Technical Novelty And Significance:** 3
**Empirical Novelty And Significance:** 3
**Recommendation:** 8

**Clarity, Quality, Novelty And Reproducibility:**

Clarity was good overall but could use some work. In particular, I had to look at ablation tables to see what data the models were pretrained on. This is very important information and should be prominent in the main text (although it's possible I missed it somewhere, please correct me if so).

The writing quality was good and the paper seems novel. It builds on VPT in a significant way.

The reproducibility is unclear but appears to be good. The authors have not provided code, and while the paper seems detailed enough to replicate, I have not attempted this.

**Strength And Weaknesses:**

##### Strengths

- Very strong results in the long-tailed regime. In particular, the Places-LT were convincing. The problem of representation learning in long-tailed scenarios is relevant quite broadly, as most natural data sources exhibit long-tailed behavior.
- Method seems intuitive (e.g. better clustering for class labels). Unclear why this leads to better generalization (see weaknesses), but the results speak for themselves.
- The method is straightforward to understand but shows creativity, in particular the strategy for integrating the group level prompt-embeddings was interesting.


##### Weaknesses

- Would appreciate more generalization results. In particular I would like to see results on all 5 ImageNet distribution shifts in Taori et al. (ImageNet-R, ImageNet-A, ImageNet-Sketch, ImageNet-V2, ObjectNet). Also how do these robustness claims fair when you pretrain on ImageNet-LT?
- In Section 5.2 (robustness), you should compare with methods which focus on robust finetuning (e.g. [1] and [2] which show good results on CLIP-pretrained backbones). It's known that full-finetuning reduces robustness but there's methods to mitigate those effects for certain classes of pretrained models. Also, does VPT exhibit similar robustness capabilities to your method? Since your LDA experiments show the features remain roughly similar, I would think it does.
- Would like to see ablation results on adding the group embeddings early vs late in layers, instead of shared prompt -> group. Would also like to see what happens if you just used group embeddings.
- You claim that "LPT shares a pretrained model for different learning tasks, and only needs to store the small-sized prompts, largely increasing the model compatibility and reducing practical deployment cost," however you don't compare to any multi-task methods. This claim is weak.
- You also make claims about the parameter efficiency of this method, however that doesn't seem to be the main selling point as you don't compare to any lightweight finetuning methods besides VPT.
- Would like to see more comparisons with CLIP-pretrained methods on iNaturalist and Places-LT. What is the result when you use your method with a CLIP pretrained backbone? I find CIFAR results to be not convincing in general, would appreciate some clarity at larger scales.

[1] Robust fine-tuning of zero-shot models, Wortsman et al. 2020
[2] Overcoming Catastrophic Forgetting by Incremental Moment Matching, Lee et al. 2017

**Summary Of The Paper:**

This paper presents a method, Long-Tailed Prompt Tuning (LPT), for light-weight finetuning of ViT's in the long-tailed image recognition regime. They identify three key issues with current methods for finetuning for long-tailed recognition:

1. Full finetuning is expensive
2. Full finetuning reduces generalization performance
3. Model features are not compatible between different finetuned models, (increased deployment cost?)

LPT attempts to fix these issues by adding a stage to the VPT training procedure, which introduces group (class) specific prompts at intermediate layers of the vision transformer. They claim that this approach makes their model have better:

1. Generalization (evaluated wrt ImageNet-Sketch)
2. Better performance in the long-tailed regime
3. Light-weight finetuning

**Summary Of The Review:**

**Update post-rebuttal:** With the promised changes to the draft, I update my review to an _accept_ (score of 8). Thank you to the authors for their thoughtful reply and overall effort.


My overall rating is _weak reject_.

The results in the long-tailed setting looks strong and the method is very interesting, however I don't think they are fleshed out enough yet for acceptance by themselves. In particular, I would like to see experiments in other long-tailed settings (e.g. the iWildCams dataset), and also for all the methods in Table 3 to be replicated in the iNaturalist setting. The multi-task claim does not appear to be fleshed out. Need results on VTAB/DomainNet for that claim to be in the paper. The generalization results could also be extended to more distribution shifts, which could provide a better understanding of the methods/improvements. You also seem to be comparing with other methods on # of tuned params. Either this is a central claim for your paper, in which case you need comparison to light-weight finetuning methods (besides VPT), or you should clarify that this is a side-benefit of your method.

My general suggestion would be to focus on the long-tailed regime and present the other experiments as side-benefits.

---

> ### Author Response · Authors · 2022-11-16
> **Author Responce to Reviewer 3, Question 1-3**
>
> ## Q1-evaluation on more OOD datasets
> Thanks for your advice. Accordingly, we further compare our LPT with the baselines (linear probe, fully fine-tune and VPT) on six different OOD datasets (ImageNet-Sketch, ImageNet-ReaL, ImageNet-V2, ImageNet-A, ImageNet-R, ObjectNet). For fairness, all the methods start from the same IN21K pretrained ViT-B and are fine-tuned on the same ImageNet-LT training set. The evaluation results are shown in the table below. It can be seen that our LPT surpasses all baselines on all the six OOD datasets. Note that as also mentioned in [5] that the pretrained ViT we used in this experiment performs poorly on ObjectNet benchmark (i.e., 17.36% accuracy after ImageNet-1k training), all results on ObjectNet in the table are relatively low because the training is performed on the subset of ImageNet-1k (i.e., ImageNet-LT). We will update these results in the revision and include this table in Appendix C.4.
>
> | Method | ImageNet-Sketch | ImageNet-ReaL | ImageNet-V2 | ImageNet-A | ImageNet-R | ObjectNet |
> | :----  | :----: | :----: | :----: | :----: | :----: | :----: |
> | Linear Probe | 31.55% | 81.43% | 63.54% | 29.20% | 45.72% | 6.61% |
> | Fully Fine-tune | 32.25% | 80.10% | 62.31% | 30.12% | 43.14% | 7.13% |
> | VPT | 34.64% | 85.82% | 68.51% | 35.17% | 47.06% | 8.03% |
> | LPT (Ours) | 36.22% | 87.22% | 70.71% | 39.65% | 50.47% | 8.22% |
>
>
> ## Q2-comparison with robust fine-tuning
> Our LPT also obtains better performance than the state-of-the-art robust fine-tuning method [4]. We conduct evaluations on six OOD datasets to verify its robustness. For fairness, both LPT and [4] start from the same IN21K pretrained ViT-B model, and are optimized on ImageNet-LT training set. The results are shown in the table below. Our LPT performs better than [4] on all OOD datasets. Besides, we also illustrate the results from vanilla VPT, which are shown in the table of R3Q1. VPT performs better than linear probing and fully fine-tuning, and performs better than [4] on 2 OOD datasets. We will update these results in the revision and include this table in Appendix C.4.
>
> | Method | ImageNet-Sketch | ImageNet-ReaL | ImageNet-V2 | ImageNet-A | ImageNet-R | ObjectNet |
> | :----  | :----: | :----: | :----: | :----: | :----: | :----: |
> | WISE-FT [4] | 34.79% | 82.20% | 65.76% | 36.75% | 47.32% | 8.00% |
> | LPT (Ours) | 36.22% | 87.22% | 70.71% | 39.65% | 50.47% | 8.22% |
>
> ## Q3-ablation study of group prompts
> We additionally conduct  two groups of ablation studies about group prompts in LPT, including early or late adding group prompts as well as training group prompts with or without shared prompts in LPT.
>
> We first investigate adding the group embeddings into early or late layers, which is shown in Table 13 of the manuscript. Note that K means the first K transformer blocks using only shared prompts to compute features. According to the results, adding group prompts into the late layers (i.e., K=8 in Table 13) leads to worse results (49.77% vs 50.07%). Adding group prompt too early (i.e., K=4) obtains nearly the same performance with LPT (49.92% vs 50.07%), but introducing more parameters (1.23M vs 0.92M). Therefore, we select K=6 as the final design of LPT. We have included these results in Appendix C.7.
>
> | K | 8 | 6 | 4 |
> | :---- | :----: | :----: | :----: |
> | Param Num | 0.61M | 0.92M | 1.23M |
> | Overall Acc | 49.77% | 50.07% | 49.92% |
>
> Second, we also investigate the effect of only optimizing group prompts, which is shown in Table 10. Only using group prompts indicates that we use only the pretrained ViT without any prompt to generate query to select matched group prompts. According to Table 10, LPT performs better than LPT without shared prompts. Although the group prompts managed to extract some domain-specific knowledge from training data after removing shared prompts (thus obtaining promising but lower results), the lack of shared prompts affects the accuracy of query selection (which is verified by the K-NN accuracy), resulting in performance degeneration.
>
> | Method | K-NN Acc | Overall Acc |
> | :---- | :----: | :----: |
> | w/o shared prompt | 32.11% | 49.81% |
> | w/ shared prompt | 36.16% | 50.07% |

---

> > ### Author Response · Authors · 2022-11-16
> > **Author Response to Reviewer 3, Question 4-6**
> >
> > ## Q4-comparison with multi-task learning framework
> > Without loss of generality, a classical multi-task learning framework consists of a shared feature extractor and multiple decoders (e.g., classification head for long-tailed classification) with each decoder used to classify samples from a specific dataset. During training, training data from all datasets are used to optimize both feature extractor and all decoders.
> > Here we compare our LPT with such a multi-task training framework from IN21K pretrained ViT-B to optimize both Places-LT and CIFAR100-LT with an imbalanced factor of 200 (IF=200). Generally, in the multi-task training method, for each task (corresponding to a specific dataset), we initialize a linear classifier, and then optimize all linear classifiers and the pretrained backbone by end-to-end fine-tuning. The corresponding results are shown in the table below. LPT surpasses multi-task training method by 10.6% on Places-LT and 14.8% on CIFAR100-LT (IF=200). Benefiting from the two merits mentioned in the introduction, LPT can achieve high performance while being easy to deploy in different scenarios. We will update these results in the revision and include this table in Appendix C.11.
> >
> > | Method | Places-LT Acc | CIFAR100-LT (IF=200) Acc |
> > | :---- | :----: | :----: |
> > | Multi-task | 39.5% | 73.1% |
> > | LPT (Places-LT) | 50.1% | - |
> > | LPT (CIFAR) | - | 87.9% |
> >
> > ## Q5-about other efficient tuning methods
> > Thanks for your suggestion. Without loss of generality, we compare LPT with more efficient tuning methods on Places-LT dataset, including different prompt tuning methods (visual prompting [1] and VPT) and adapter tuning methods (e.g., Pro-Tuning [2]). We also additionally adopt two efficient tuning methods, i.e., normalization tuning [3] and adapter tuning into our LPT (i.e., LPT+ft-norm and LPT+Adapter).
> > As for previous efficient tuning methods, we adopt balanced sampling to optimize them.
> > Without loss of generality, we keep using the same IN21K pretrained ViT as backbone for all methods and evaluate all of them on Places-LT dataset.
> >
> > The evaluation results are shown in the following table. It can be seen that our LPT beats all the baselines, e.g. 5.9% and 1.8% higher than the state-of-the-art VPT and Pro-Tuning, respectively, demonstrating its effectiveness on handling long-tailed classification. Note that [1] only adds learnable prompt on the edges of input images, and fixes all the other pretrained parameters (including the linear classifier), which may explain its relatively lower performance.  Meanwhile, after adopting other efficient tuning methods into LPT, the final performance has no significant improvement. Thus we choose the current LPT structure as the final framework. We will update these results in the revision and include this table in Appendix C.3.
> > | Method | Overall Acc |
> > | :---- | :----: |
> > | VPT | 44.2% |
> > | [1] | 13.8% |
> > | Pro-Tuning | 48.3% |
> > | LPT | 50.1% |
> > | LPT+ft-norm | 49.9% |
> > | LPT+Adapter | 49.9% |
> >
> > ## Q6-about CLIP-based methods
> > There are only three existing methods (i.e., BALLAD, VL-LTR and RAC) using CLIP-pretrained models to handle long-tailed classification, and we compare with all of them. See Tables 3-5 in the paper for details.
> >
> > However, we claim that using CLIP-pretrained backbone (especially using extra data during testing) to conduct long-tailed learning for comparison is unfair for previous state-of-the-art methods, as CLIP uses enormous 400M image-text pairs to optimize the model and its text encoder can provide robust classification information to image encoder, which is nearly infeasible for previous methods with ImageNet-pretrained backbones. Nevertheless, without using extra wiki text data or IN21K database during testing, LPT still surpasses CLIP-pretrained BALLAD on multiple datasets, which is shown in Tables 3 and 4 in the paper.
> >
> > Besides, the reason why we additionally evaluate performance on CIFAR100-LT is that CLIP-pretrained BALLAD evaluated the performance on CIFAR100-LT dataset. To make comparison with BALLAD, except for Places and iNaturalist, we also evaluate LPT on CIFAR100-LT. Using three benchmarks rather than two can make the evaluation more comprehensive.
> >
> >
> > [1] Exploring Visual Prompts for Adapting Large-Scale Models, arxiv 2022
> > [2] Pro-tuning: Unified prompt tuning for vision tasks, arxiv 2022
> > [3] Training batchnorm and only batchnorm: On the expressive power of random features in cnns, ICLR 2021
> > [4] Robust fine-tuning of zero-shot models, CVPR 2022
> > [5] Pyramid Adversarial Training Improves ViT Performance, CVPR 2022

---

> > ### Comment · Reviewer_mA6r · 2022-11-21
> > **Great rebuttal, some clarifications requested**
> >
> > The experiments included here are excellent, and I believe make your overall paper stronger. Can you clarify your robustness experiments against Wise-FT? What $\alpha$ mixing coefficent do you use between the zero-shot and finetuned model?
> >
> > Your numbers for ObjectNet are significantly worse than even a model trained on ImageNet, let alone ImageNet21k + ImageNet-LT. What is it about ImageNet-LT that would harm results on ObjectNet specifically? Can you share what your models get on the ImageNet-1k val set?
> >
> > Finally, I understand why you want to pretrain everything on ImageNet-21k for a similar comparison. However, CLIP pretrained models are ubiquitous and have interesting robustness properties. Would it be possible to repeat your robustness experiments with a CLIP pretrained backbone? I do not require it for raising my score, however I believe it would make your paper stronger, as it would allow direct comparison with other robustness works.
> >
> > Overall, great rebuttal. I will raise your score to an 8.

---

> > > ### Author Response · Authors · 2022-11-23
> > > **Re: Great rebuttal, some clarifications requested**
> > >
> > > Sincerely thanks for your suggestion. Here we provide more details to clarify the rebuttal.
> > >
> > > ## Q1-details of WISE-FT
> > >
> > > Here we clarify the details of WISE-FT with ViT pretrained on ImageNet-21K. For the fully fine-tuned model in WISE-FT, we follow the main idea of WISE-FT and fully fine-tuned the pre-trained ViT on ImageNet-LT train set as fully fine-tuned model. And for zero-shot model, different from zero-shot CLIP where the CLIP text encoder can be seen as a classifier, we conduct linear probing on ImageNet-LT train set to obtain an optimal classifier and use the linear probing ViT as zero-shot model. Both models start from the same ViT-B with ImageNet-21K pretrained weights. During WISE-FT, we follow [4] and conduct grid search on ImageNet-A dataset, where α=0.1,0.2,...,0.9. Finally we find that WISE-FT with α=0.5 achieves the best accuracy, thus we evaluate all OOD datasets on WISE-FT with α=0.5.
> > >
> > > ## Q2-the reasons of low ObjectNet scores
> > >
> > > The low ObjectNet scores come from two fatal components.
> > >
> > > First, the long-tailed training set (i.e., ImageNet-LT) significantly harms the results on ObjectNet score. To verify this point, we use the same evaluation code to evaluate the official ImageNet-1K pretrained ResNet-50 released by PyTorch [3] as well as the ImageNet-LT pretrained ResNet-50 released by [1] on ObjectNet dataset. Both two pretrained models use the exactly same training schedule to obtain the final pretrained weights. Following the official code of ObjectNet [2], we directly evaluate the accuracy on the whole ObjectNet val set. And for fairness, we also report the ObjectNet accuracy on 113 overlapping classes with ImageNet-1K. The evaluation results are shown in the following table. We find that, using long-tailed dataset (i.e., ImageNet-LT) leads to a large performance degeneration on both ImageNet-LT val dataset as well as ObjectNet dataset. A feasible explanation is that, for ImageNet-LT pretraining (e.g., [1]) or prompt tuning (e.g., our LPT), even if introducing advanced training techniques (e.g., [1] or our LPT), it is still difficult to obtain optimal classifier because the huge imbalance between many-shot classes and few-shot classes. Thus the sub-optimal classifier generate relatively low-quality prediction results on samples from ObjectNet, obtaining low accuracy. We will release all evaluation code we used for reproducing the evaluation results.
> > >
> > >
> > > | Model | train set | ImageNet Acc | ObjectNet Acc | ObjectNet Acc (113 overlapping classes) |
> > > | :---- | :----: | :----: | :----: | :----: |
> > > | ResNet-50 | ImageNet-1k | 76.13% | 3.73% | 10.20% |
> > > | ResNet-50 | ImageNet-LT | 43.13% | 0.99% | 2.71% |
> > >
> > > Second, the inherent large domain shift (e.g., rotation, background, viewpoint) also leads to low evaluation scores on ObjectNet dataset, which has been thoroughly discussed in [2]. Therefore, the low accuracy on ObjectNet scores may also be explained by the large domain shift from ImageNet to ObjectNet.
> > >
> > > ## Q3-about robustness experiments with CLIP-pretrained backbone
> > >
> > > Thanks for your comments. Now we all agree that using ImageNet-21K pretrained weights rather than CLIP-pretrained weights can make fair comparison with previous long-tailed learning methods. For completeness, we will conduct corresponding robustness experiments with CLIP-pretrained backbone and update these results in the revision. We believe that these results can make our paper stronger and allow direct comparison with other robustness works.
> > >
> > > [1] Decoupling Representation and Classifier for Long-Tailed Recognition, ICLR 2020
> > >
> > > [2] ObjectNet: A large-scale bias-controlled dataset for pushing the limits of object recognition models, NeurIPS 2019
> > >
> > > [3] PyTorch: An Imperative Style, High-Performance Deep Learning Library, NeurIPS 2019
> > >
> > > [4] Robust fine-tuning of zero-shot models, CVPR 2022

---

### Official Review · Reviewer_ferv · 2022-10-25

**Confidence:** 3
**Correctness:** 3
**Technical Novelty And Significance:** 2
**Empirical Novelty And Significance:** 3
**Recommendation:** 6

**Clarity, Quality, Novelty And Reproducibility:**

The article is well-written, the problem statements are clear, the conclusions are supported by experimental results, and there are no grammatical errors in the writing aspect.
The core idea and components of this method exist in the previous research, so its innovation is not that outstanding.


**Strength And Weaknesses:**

Strengths:
1.	Compared with previous work, the LPT method can effectively handle various long-tailed learning scenarios by fine-tuning fewer parameters, and the model has strong generalization ability and compatibility.
2.	The experimental settings are reasonable, and the authors have done relatively complete ablation experiments.

Weaknesses:
1.	Compared with the VPT method, the LPT method only adds a small number of fine-tuning parameters, but the manuscript does not mention whether the 2-stage method will increase the training time.
2.	The LPT method is robust to the domain shift problem. This paper proposes a possible explanation, but does not specify that domain-specific knowledge is obtained by what kind of prompt and at which stage.


**Summary Of The Paper:**

This submission focuses on the long-tailed classification tasks, and proposes the LPT method to alleviate the negative problems of long-tailed learning. The training of LPT consists of two stages. The first stage trains shared prompts to learn general features of the dataset, and the second stage trains group-specific prompts to learn specific features for samples with similar features. The contribution of this paper is to apply the idea of VPT to the long-tailed classification problem and achieves effective improvement.

**Summary Of The Review:**

The authors are inspired by VPT, like using prompts to fine tune the network, and apply this kind of problem-solving idea to the study of long-tailed data. Although this idea exists in previous research, it is undeniable that this method has achieved some effective improvement in experimental performance, so I think it is a meaningful work.

---

> ### Author Response · Authors · 2022-11-16
> **Author Responce to Reviewer 2**
>
> ## Q1-about training cost
> Actually, LPT is more training-efficient than previous methods. Without loss of generality, as an example, we use the Places-LT dataset to evaluate the overall training cost of the model [1], which uses traditional 100 epochs training schedule. The evaluation is done on a single RTX A6000 GPU with batch size of 256. We first evaluate the average training time per epoch for each phase of LPT and [1], with results summarized in the following table. It can be seen that LPT (Phase 1) uses the same training time with [1], while LPT (Phase 2) uses shorter training time for each epoch. These results indicate that LPT does not lead to increased training time for each training epoch.
>
> | Method | Training Time (seconds / epoch) |
> | :----  | :----: |
> | LPT (Phase 1) | 219.6  |
> | LPT (Phase 2) | 209.8  |
> | [1] | 220.5  |
>
> Then, we compute the total training time of both methods based on per-epoch training time and corresponding training schedules reported in original literature. For [1], the overall training time is 220.5 * 100 / 3600 = 6.13 hours, while LPT only uses (219.6 * 40 + 209.8 * 40) / 3600 = 4.77 hours. Therefore, it is again shows that LPT does not bring increased overall training time.
>
> ## Q2-about domain-specific knowledge
> As discussed in Sec. 3 and Sec 4.1, the domain-specific knowledge is obtained by shared prompts in Phase 1 of LPT. To further verify this point, we also adopt LDA among features extracted from LPT (Phase 1) and LPT (Phase 2), as well as features from pretrained ViT without any prompts. The visualization results are provided in Appendix C.1 of the revision. According to the visualization, we find that the distributions of features from LPT (Phase 1) and LPT (Phase 2) are highly overlapped, showing that shared prompts in Phase 1 mainly obtain domain-specific knowledge. Moreover, we also investigate the γ value and inter-class distance of features from LPT (Phase 1) and LPT (Phase 2). We find that both of them have similar γ values, while LPT (Phase 2) has larger inter-class distance, indicating that group prompts bring more class discriminative ability.
> |   Metric  | Phase 1 | Phase 2 |
> | :---- | :----: | :----: |
> | Inner-class / Inter-class γ | 0.237 | 0.242 |
> | Inter-class distance | 15.8 | 16.1 |
>
> [1] Decoupling Representation and Classifier for Long-Tailed Recognition, ICLR 2020

---

### Official Review · Reviewer_qoSe · 2022-10-25

**Confidence:** 4
**Correctness:** 3
**Technical Novelty And Significance:** 3
**Empirical Novelty And Significance:** 3
**Recommendation:** 6

**Clarity, Quality, Novelty And Reproducibility:**

Overall, the paper is mainly focused on the design of phase 2, but according to the ablation in Table 8, the contribution of phase 2 is minor.

1. In table 3, the column of tuned parameters is misleading, even though it is right. Most of the previous methods are learned from scratch, however, the proposed method is based on a pretrained model on much larger datasets, which is unfair.

2. what is the case of type (a) in the ablation "Effect of Each Phase"? How to understand the case with phase I but without prompt?

3. In section 4.3, the introduction and motivation of GCL loss seem missing.

**Strength And Weaknesses:**

Strength:

1. It's a good try to explore prompt tuning for long-tailed recognition.
2. The experiments and related analyses are comprehensive and clear.
3. The paper is well-written and easy to follow.

Weaknesses:

1. The biggest concern is the leverage of pretrained models, which are based on much larger datasets. This will make it unfair to compare with previous baselines. This point should be discussed.

2. Since most of previous methods are based on resnet. The difference between different backbones should also be discussed. The difference should be shown clearly in the table.

Thus, in all tables in the experiments, it is suggested to provide the corresponding backbone and valid parameters of the proposed method and baselines for a fair comparison.

3. Why the many-shot accuracy of VPT is substantially below than linear probing under balanced sampling in table 1?
4. The adoption of pretrained model leads to the case that the method is naturally more advantageous than previous methods.  Thus, a more comprehensive comparison between different prompt learning methods is needed in this setting. These methods can be those discussed in the related works.

**Summary Of The Paper:**

This paper introduces an effective prompt-tuning method for long-tailed recognition. The proposed method consists of two phases. They first learn the shared prompt and then consider group prompts. They conduct experiments on Places-LT, Cifar100-LT and iNaturalist 2018. The results are good and comprehensive, which validates the efficacy of the proposed method.


**Summary Of The Review:**

This paper proposes a new prompt-tuning based method for tackling with long-tailed datasets. The results and the analysis are good. However, there are some points which need further clarification.

---

> ### Author Response · Authors · 2022-11-16
> **Author Response to Reviewer , Question 1-3**
>
> ## Q1-about pretrained models with larger datasets.
>
> Thanks for your question. For training on Places-LT and iNaturalist, several previous works also leveraged IN1k pretrained models (e.g., IN1k pretrained Res152) in their experiments. To fairly compare with them, we conduct two groups of experiments:
>
> First, we analyze LPT with a same vision backbone (i.e., ViT-B) but using different pretraining data scale (including IN1K and IN21K) and different pretraining algorithms (including supervised pretraining, DINO and Mugs), and report the results on Places-LT dataset in the following table (also Table 11 in main paper).
> One can observe that on Places-LT dataset, LPT with IN1K supervised pretrained weights achieves 42.7% overall accuracy, and improves the state-of-the-art PaCo (41.2% overall accuracy) by 1.5%. Meanwhile, with IN1K Mugs pretrained weights, LPT further achieves 43.66% overall accuracy on Places-LT dataset. These results indicate that, even using the same pretraining data (i.e., IN1k) to initialize the backbone, LPT still obtains better performance. We have included these results in Appendix C.5, and will update these results in the revision.
>
> | Method | Data | Overall Acc |
> | :----  | :----: | :----: |
> | DINO   | IN1K   | 42.18% |
> |  Mugs  | IN1K   | 43.66% |
> | Supervised | IN1K | 42.71% |
> | Supervised | IN21K | 50.07% |
>
> Second, to fairly compare LPT with previous methods, we reimplement three state-of-the-art long-tailed methods (i.e., Kang et al. (2020), PaCo (2021) and GCL (2022)) with the same backbone and the same IN21K pretrained ViT-B as feature extractor to conduct fully fine-tuning on Places-LT.
> The following table summarizes the experimental results. It can be seen that LPT surpasses all baselines by a large margin. Especially, it beats the state-of-the-art GCL (2022) with IN21K pretrained ViT-B backbone, which achieves 42.82% overall accuracy on Places-LT, i.e. 7.25% lower than our LPT. We will update these results into the revision and include this table in Appendix C.2.
>
> | Method | Backbone | Data | Overall Acc |
> | :---- | :----: | :----: | :----: |
> | Kang et al. (2020) | ViT-B | IN21K   | 40.45% |
> |  PaCo (2021) | ViT-B | IN21K   | 37.00% |
> | GCL (2022) | ViT-B | IN21K | 42.82% |
> | LPT | ViT-B | IN21K | 50.07% |
>
> ## Q2-clarify the statement of Table 3 (main results about Places-LT)
> Thanks for your suggestion. We will update the description of Table 3 in the revision. We will also add the reproduced results (shown in below table) of previous state-of-the-art methods using IN21K pretrained ViT-B backbone into Appendix C.2.
>
> | Method | Backbone | Data | Overall Acc |
> | :---- | :----: | :----: | :----: |
> | Kang et al. (2020) | ViT-B | IN21K   | 40.45% |
> |  PaCo (2021) | ViT-B | IN21K   | 37.00% |
> | GCL (2022) | ViT-B | IN21K | 42.82% |
> | LPT | ViT-B | IN21K | 50.07% |
>
>
> ## Q3-why many-shot accuracy of VPT is lower
> Thanks for your question. The reason why many-shot accuracy of VPT training with balanced sampling is lower than linear probing is that, the pretrained backbones are usually optimized by large-scale many-shot training data, and some many-shot classes (e.g., restaurant, valley and palace) from downstream long-tailed datasets may exist in the pretraining dataset. Therefore, for these many-shot samples, the pretrained model may obtain better feature representation than that for other medium/few-shot samples. In this way, a linear classifier can easily recognize such many-shot samples, and may perform poorly on other classes. Different from linear probing, VPT can fine-tune an additional prompt to handle the domain shift for the long-tailed training data. However, since balanced sampling is used during fine-tuning, few-shot samples may be utilized more than many-shot samples during training. Thus the learned prompt tends to fit better on few-shot classes, and impairs the fitting performance on many-shot classes. This phenomenon may restrict the discriminative ability of corresponding linear classifier in VPT on many-shot classes. Therefore the many-shot accuracy of VPT is lower than linear probing.

---

> > ### Author Response · Authors · 2022-11-16
> > **Author Response to Reviewer 1, Question 4-7**
> >
> > ## Q4-more comprehensive comparison between different prompt learning
> >
> > Thanks for your suggestion. Without loss of generality, we conduct comparisons between LPT and more efficient tuning methods on Places-LT dataset, including different prompt tuning methods (visual prompting [1] and VPT) and adapter tuning methods (e.g., Pro-Tuning [2]). Furthermore, we also additionally adopt two efficient tuning methods, i.e., normalization tuning [3] and adapter tuning into our LPT (i.e., LPT+ft-norm and LPT+Adapter).
> > As for previous efficient tuning methods, we adopt balanced sampling to optimize the models.
> > Without loss of generality, we keep using the same IN21K pretrained ViT as backbone for all methods and evaluate all them on Places-LT dataset.
> >
> > The corresponding evaluation results are shown in the following table. It can be seen that our LPT beats all the baselines, e.g. 5.9% and 1.8% higher than the state-of-the-art VPT and Pro-Tuning, respectively, demonstrating its effectiveness on handling long-tailed classification. Note that [1] only adds learnable prompt on the edges of input images, and fixes all the other pretrained parameters (including the linear classifier), which may explain its relatively lower performance.  Meanwhile, after adopting other efficient tuning methods into LPT, the final performance has no significant improvement. Thus we choose the current LPT structure as the final framework. We will update these results in the revision and include this table in Appendix C.3.
> > | Method | Overall Acc |
> > | :---- | :----: |
> > | VPT | 44.2% |
> > | [1] | 13.8% |
> > | Pro-Tuning | 48.3% |
> > | LPT | 50.1% |
> > | LPT+ft-norm | 49.9% |
> > | LPT+Adapter | 49.9% |
> >
> > ## Q5-about training from scratch
> > To provide more accurate information in Table 3, we will update the description of Table 3 (e.g., adding backbone and its parameter amount information) in the revision.
> >
> > For a fair comparison with previous methods in Table 3, all previous methods trained on Places-LT and iNaturalist datasets are optimized from ImageNet-pretrained weights.
> > We also adopt IN21K pretrained backbone into three different long-tailed learning methods and evaluate the overall accuracy, which is reported in the reply to R1Q1. Our LPT surpasses all these methods on Places-LT.
> >
> > ## Q6-about ablation study
> > For visual prompt tuning methods (including LPT), both prompts and the additional linear classifier are fine-tuned during training. Specifically, in type (a) setting the shared prompts are removed, with the rest strictly following type (b) setting, i.e. Phase 1 of LPT. From type (a) results in Table 8 of the manuscript, it can be seen that using shared prompts can also lead to significant performance improvement, as it also helps handle domain shift between pretrained models and target long-tailed datasets.
> >
> > ## Q7-about motivation of GCL loss
> > The motivation of introducing Gaussian clouded logit (GCL) loss (Li et al. 2022) into our LPT is as follows. Each data can be seen as a Gaussian distribution in the whole data space. For highly imbalanced training data, to simulate the ideal balanced data space, samples from many-shot classes should have a smaller variance (boundary) to restrict the boundary of each many-shot class, while those from few-shot classes should obtain a larger boundary to cover more space to simulate the distribution of many-shot classes. Therefore, GCL loss is used to adaptively set the variance of each training sample for better long-tailed learning, which is crucial to LPT.
> > Hence, in LPT we start from GCL loss and propose A-GCL loss for LPT training.
> >
> > [1] Exploring Visual Prompts for Adapting Large-Scale Models, arxiv 2022
> > [2] Pro-tuning: Unified prompt tuning for vision tasks, arxiv 2022
> > [3] Training batchnorm and only batchnorm: On the expressive power of random features in cnns, ICLR 2021

---

> > > ### Comment · Reviewer_qoSe · 2022-11-25
> > > **Response to rebuttal**
> > >
> > > I think the authors have well-addressed my concerns.
> > >
> > > However, should not a revised version with highlights be updated to the system to reveal the changes?
> > >
> > > Thank you so much.

---

> > > > ### Author Response · Authors · 2022-11-25
> > > > **Re: Response to rebuttal**
> > > >
> > > > Sincerely thanks for your reply. In the first rebuttal phase, we have uploaded the revised pdf. To better clarify the modification, we provide the details of the revision as follows:
> > > >
> > > > 1) For question 1 (i.e. pretrained models with larger datasets), we add Appendix C.2 and Table 14 (in page 16) to fairly compare our LPT with previous SoTA methods. For LPT and SoTAs, they all use ViT-B backbone pretrained on ImageNet-21K. But LPT still outperforms SoTA baselines.
> > > >
> > > > 2) For question 2 (clarify the statement of Table 3 in page 7), we add two columns (i.e., Backbone and Total Params) on Table 3 to clarify the statement of all methods and the comparison with previous SoTA methods.
> > > >
> > > > 3) For question 4 (more comprehensive comparison between different prompt learning), we add Appendix C.3 and Table 15 (on page 16) to provide more comparison results among previous efficient tuning methods, including prompt tuning, adapters, and other efficient tuning techniques.
> > > >
> > > > 4) For question 2 from Reviewer ferv (about domain-specific knowledge), we add Appendix C.1, Fig. 5, and Table 13 (on page 15) to discuss the effect of shared prompt and group prompts in our LPT.
> > > >
> > > > 5) For questions 1 and 2 from Reviewer mA6r (more detailed robustness evaluation), we add Appendix C.4 and Table 17 (on page 17) to make a comparison between LPT and four different fine-tuning strategies on six different OOD datasets.
> > > >
> > > > 6) For question 4 from Reviewer mA6r (comparison with multi-task learning), we add Appendix C.11 and Table 21 (on page 19) to make the comparison between LPT and a classical multi-task learning method on two long-tailed datasets (i.e., Places-LT and CIFAR100-LT with IF=200).
> > > >
> > > > Sincerely sorry for the confusion caused by not highlighting the modification. We hope this clarification can help you read the modification of the revision. Many thanks.

---

> > > > > ### Comment · Reviewer_qoSe · 2022-11-25
> > > > > **Thanks for the immediate reply**
> > > > >
> > > > > It is actually just a suggestion to make the corresponding changes easier to capture.
> > > > >
> > > > > I raise my score to 6.

---

### Public Comment · ~Wenhai_Wan1 · 2023-06-06
**what's the differences between Phase 1 and VPT?**

as the title described.

---

> ### Author Response · Authors · 2023-06-06
> **Re: what's the differences between Phase 1 and VPT?**
>
> The difference is the training strategy.
> "VPT" indicates the vanilla visual prompt tuning without any training technique for long-tailed learning.
> "Phase 1" includes both shared prompts and the specially-designed training strategy for long-tailed learning (e.g., dual-sampling, loss functions, weight-decay, etc.). Above two lines in ablation study indicates that, even using the vanilla shared prompt like VPT, with specially-designed training strategy, the performance can still obtain promising improvement.

---

### Decision · Program_Chairs · 2023-01-20

**Decision:**

Accept: poster

**Justification For Why Not Higher Score:**

The problem that the paper addresses would interest somewhat of a narrow community. The idea wont necessarily generalizes to other subcommunities in ICLR.

**Justification For Why Not Lower Score:**

Good simple idea, strong results, all reviewers recommended accept.

**Metareview: Summary, Strengths And Weaknesses:**

This paper addresses fine-tuning of ViT's in the setup of transfer learning for image recognition with long-tailed data.
Their analysis of current fine-tuning approaches suggest that it is expensive and may hurt generalization of a pre-trained model.
To address this, they propose a prompt-tuning method, for shared prompts and group-specific prompts.
Strong empirical results on Places-LT, CIFAR-LT

Reviewers had various proposals to strengthen the paper, including additional experiments,
which the reviewers addressed in the rebuttal. All four reviewer recommended accepting the paper.

As a reviewer noted, this paper focuses on "transfer pre-trained knowledge to unbalanced data". This has a differnt scope than
the more usual " learn from imbalanced data", which makes it depend on how balanced the foundation model is.

The paper's main strength is in the empirical evaluation, showing significant improvement of accuracy over earlier approaches.


**Note From Pc:**

if the above contains the word "oral" or "spotlight" please see: "oral" presentation means -> notable-top-5% and "spotlight" means -> notable-top-25%. As stated in our emails, we are disassociating presentation type from AC recommendations